

# SIZE DEPENDENT CHEMICAL AGEING OF OLEIC ACID AEROSOL UNDER DRY AND HUMIDIFIED CONDITIONS

**Suad Al-Kindi, Francis D. Pope\*, David C. Beddows[‡],**

**William J. Bloss and Roy M. Harrison[†‡]**

**School of Geography, Earth and Environmental Sciences**

**University of Birmingham**

**Edgbaston, Birmingham B15 2TT**

**United Kingdom**

20

\* Corresponding author: f.pope@bham.ac.uk
[†] Also at: Department of Environmental Sciences / Center of Excellence in Environmental Studies, King Abdulaziz University, Jeddah, 21589, Saudi Arabia
[‡] Also at: National Centre for Atmospheric Science, United Kingdom



**ABSTRACT**
A chemical reaction chamber system has been developed for the processing of oleic acid aerosol
particles with ozone under two relative humidity conditions: dry and humidified to 65% R.H..  The
apparatus consists of an aerosol flow tube, in which the ozonolysis occurs, coupled to a scanning
mobility particle sizer (SMPS) and an aerosol time-of-flight mass spectrometer (ATOFMS) which
measure the evolving particle size and composition. Under both relative humidity conditions,
ozonolysis results in a significant decrease in particle size and mass which is consistent with the
formation of volatile products that partition from the particle to the gas phase.  Mass spectra derived
from the ATOFMS reveal the presence of the typically observed reaction products:  azaleic acid,
nonanal, oxononanoic acid and nonanoic acid, as well as a range of higher molecular weight
products deriving from the reactions of reaction intermediates with oleic acid and its oxidation
products.  These include octanoic acid, and 9- and 10-oxooctadecanoic acid, as well as products of
considerably higher molecular weight.  Quantitative evaluation of product yields with the ATOFMS
shows a marked dependence upon both particle size association (from 0.3 to 2.1 µm diameter) and
relative humidity.  Under dry conditions, the percentage residual oleic acid increases with
increasing particle size, as does the percentage of higher molecular weight products, due to the
poorer internal mixing of the larger particles.  The main lower molecular weight products are
nonanal and oxononoic acid.  Under humidified conditions, the percentage unreacted oleic acid is
greater, except in the smallest particle fraction, and oxononanoic acid dominates the product
distribution, with little formation of high molecular weight products relative to the dry particles.  It
is postulated that water reacts with reactive intermediates, competing with the processes which
produce high molecular weight products.  Whilst the oleic acid model aerosol system is of limited
relevance to complex internally mixed atmospheric aerosol, the generic findings presented in this
paper give useful insights into the nature of heterogeneous chemical processes.
**Keywords:** Heterogeneous reactions; oleic acid; ozone; ATOFMS



## 1.    INTRODUCTION

Atmospheric aerosol particles play critical roles in air quality, visibility, human health, regional and

global climate, cloud condensation nuclei ability, precipitation events, atmospheric acid deposition,

optical properties, atmospheric energy balance, and stratospheric ozone depletion (Harrison, 2014).

Aerosol particles are typically composed of a mixture of inorganic and organic material. The

organic component of aerosol is highly complex and may contain thousands of different chemical

species of both biogenic and anthropogenic origin (Goldstein and Galbally, 2007).  These organic

components cause aerosol particles to exhibit a wide range of chemical properties due to their

differing composition.

Primary organic aerosol (POA) particles are emitted directly from anthropogenic and biological

sources (Pöschl, 2011). It is composed of a wide range of hydrocarbons, partially oxidized organics,

and elemental carbon primarily related to combustion processes including burning of fossil fuels,

cooking, domestic heating, and biomass burning. Natural biogenic sources of organic aerosol

particles include plants and vegetation, the ocean surface, volcanic eruptions, and wind-driven dust

(Pöschl, 2011). By contrast, secondary organic aerosol (SOA) particles are formed in the

atmosphere from biogenic and anthropogenic gaseous precursors. Several pathways for the

transformation of aerosol particles in the atmosphere have been identified, which may alter their

chemical and physical properties, in general causing hydrophobic-to-hydrophilic conversion of

organic components. These pathways include gas-phase reactions, condensed-phase reactions,

multiphase reactions, and multigenerational chemistry which has received attention recently

because it is the least understood (Rudich, 2003; Petters et al., 2006; Kroll and Seinfeld, 2008;

Carlton et al., 2010; Koop et al., 2011; Kolb and Worsnop, 2012).

The ozonolysis of oleic acid (OL) aerosol is a much studied heterogeneous reaction which provides

a readily accessible test system for the understanding of atmospheric processing of organic aerosol



under a range of environmental conditions (Hearn and Smith, 2004; Katrib et al., 2004; Ziemann,
2005; Gonzalez-Labrada et al., 2007; Zahardis and Petrucci, 2007; Vesna et al., 2009; Pfrang et al.,
2010; Lee et al., 2012; Chan et al., 2013; Hosny et al., 2013; Mendez et al., 2014; Hosny et al.,
2016) - although oleic acid itself is only introduced into the atmosphere in small quantities via the
heating of fat and cooking oil.  OL studies have, for example, explored the effects of the droplet
state (Katrib et al., 2005a), relative humidity (RH) (Vesna et al., 2009; Lee et al., 2012), OL and
ozone concentration (Lee and Chan, 2007; Mendez et al., 2014),  and extent of chemical ageing
(Reynolds et al., 2006) upon the OL - ozone system. The kinetics (Moise and Rudich, 2002;
Gonzalez-Labrada et al., 2007), reaction mechanism and products (Hearn and Smith, 2004; Katrib
et al., 2004; Hung et al., 2005; Zahardis et al., 2005; Ziemann, 2005; Zahardis et al., 2006; Zahardis
and Petrucci, 2007), particle morphology and hygroscopicity (Dennis-Smither et al., 2012a), and
viscosity (Hosny et al., 2013; 2016) have also been investigated.
The initial stages in the mechanism of particulate phase OL ozonolysis are comparatively well
understood (Moise and Rudich, 2002; Smith et al., 2002; Hearn and Smith, 2004; Katrib et al.,
2004; Ziemann, 2005; Grimm et al., 2006; Nash et al., 2006; Hung and Ariya, 2007; Zahardis and
Petrucci, 2007; Vesna et al., 2008; Lee et al., 2012), see Scheme 1. The initial step in the reaction is
the addition of ozone across the double bond of OL forming an unstable primary ozonide (POZ).
Subsequently, the POZ decomposes, via two potential routes, through cleavage of the C-C bond
alongside one of the two O-O bonds. Both routes generate an aldehyde and an excited Criegee
Intermediate (CI) as products, but with differing chemical identity depending upon which O-O bond
is broken. In reaction route 1 (Scheme 1), nonanal (NN) and CI1 are formed; the CI1 can isomerise
to form stabilized azelaic acid (AA), a cyclic acyloxy hydroperoxide (CAHP), or octanoic acid
(OcA) and carbon dioxide (Gonzalez-Labrada, Schmidt et al. 2007). Alternatively, CI1 may be
scavenged via numerous potential association reactions with the co-produced aldehyde, other
carboxylic-functionalised moieties, (further) OL molecule double bonds, solvents or via self-





reaction. This extensive secondary chemistry reflects the reactivity of CIs (Zahardis and Petrucci,
2007). Similarly, in reaction route 2, oxo-nonanoic acid (ON) and excited Criegee intermediate 2
(CI2) are formed, and CI2 decomposes by forming stabilized nonanoic acid (NA), or may be
scavenged via similar pathways as described above for CI1. The principal reaction products of OL
ozonolysis are therefore NN, AA, ON, and NA.  There are also numerous reports in the literature of
the formation of higher molecular weight products (Hung et al., 2005; Reynolds, et al., 2006;
Zahardis et al., 2005) including esters (Hung et al., 2010) and peroxides (Reynolds et al., 2006;
Vesna et al., 2009; Zahardis et al., 2005; 2006; Ziemann, 2005).  Although OL is essentially
hydrophobic, the reaction products are hydrophilic (Lee et al., 2012).

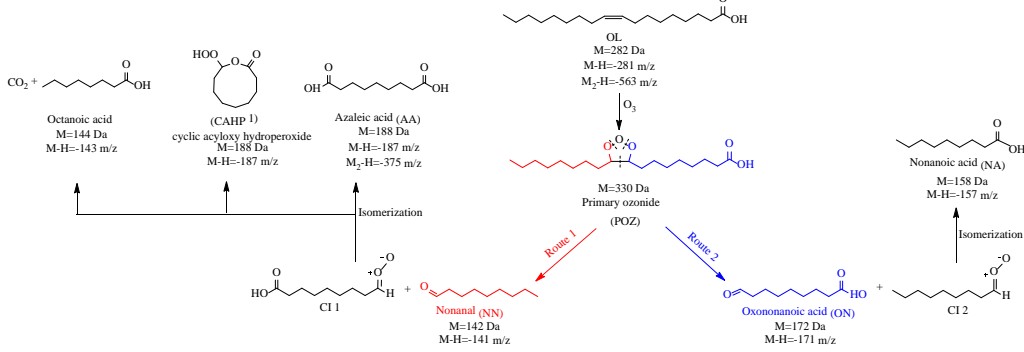

**Scheme 1:**  Initial steps and primary products of Oleic Acid (OL) oxidation by $O_3$.
This study investigates the chemical mechanisms of OL ozonolysis and the resulting product
distribution, assessed as a function of particle size using a variety of novel online physical and
chemical analysis methodologies.  The effect of relative humidity upon the (size dependent)
predominant reaction mechanism is also explored, and the resulting implications for the
atmospheric processing of organic aerosol are considered.



**2.    EXPERIMENTAL**
**2.1    Experimental Design**
A schematic diagram of the experimental setup for the ozonolysis of OL aerosol is shown in Figure
1. The system comprised a gas handling and control system, an aerosol generator, ozone generator,
humidifier (bubbler), and an aerosol flow tube (AFT), coupled to aerosol particle characterisation
(SMPS - Scanning Mobility Particle Spectrometer - and ATOFMS - Aerosol Time-of-Flight Mass
Spectrometer) instruments, and monitors to measure ozone, temperature and relative humidity
(RH). The setup was designed so the ozonolysis of OL aerosol could be studied under well-defined
conditions within the AFT.
The AFT reactor consisted of a cylindrical tube formed from Pyrex (internal diameter = 10 cm,
length = 100 cm) which was sealed with aluminium flanges. The reactor was held in a vertical
orientation within a supporting frame. Gas inlets and outlets were coupled to the flanges to allow
for the entrance and exit of the gas and aerosol streams. The ozone monitor, SMPS and ATOFMS
sampling were located immediately downstream of the AFT. Typical gas flows entering the AFT
were as follows : OL aerosol containing flow (1.0 slm in synthetic air), ozone containing dry
synthetic air (0.5 slm), and dry or humidified synthetic air (2.0 slm) resulting in a total flow rate of
3.5 slm which corresponded to a (plug flow) residence time of 135 s. In some experiments a
nitrogen flow was used to generate the aerosol (via homogeneous nucleation of OL); in these cases
a small flow of oxygen was added to ensure the air composition within the AFT had an atmospheric
$N_2:O_2$ ratio. Constant gas flows were achieved using calibrated mass flow controllers. The
Reynolds number for the AFT was 13.1 and hence flow within the reactor was predicted to lie
firmly within the laminar regime after an initial flow development length (estimated at 4.6 cm, i.e. <
5% of the total length).





After passing through the AFT aerosol particles were analysed using the SMPS and ATOFMS
instruments.  The gaseous ozone concentration was also measured at the AFT exit; tests were
performed to verify that the ozonolysis reaction did not significantly change the ozone
concentration.  All tubing (external diameter = 6 mm) and connections were made of either PTFE or
stainless steel (Swagelok Manchester Fluid) except for those used to introduce or monitor the
aerosol which was of antistatic construction (Grimm Aerosol).
To ensure that particle processing was dominated by the AFT section of the setup, the additional
residence time within the sampling and characterisation stages was assessed. The calculated
residence time of the aerosol in the sampling tubes (<2 s) and SMPS (<1 s) (von Hessberg et al.,
2009) was negligible. Within the ATOFMS the residence time is very small.  However, the low
operating pressures within the ATOFMS (~$10^{-7}$ Torr) minimize interactions between ozone and
particles, and hence particle reactivity is considered to be terminated at the very early stages of the
ATOFMS sampling nozzle.
Elevated RH within the AFT was generated by passing a 2.0 slm flow of synthetic air through two
water sequential bubblers containing deionized water, prior to entering AFT. A hygrometer probe
(Rotronic, hygropalm) positioned inside the AFT was used to monitor RH and temperature.
Experiments were typically run under both dry and humidified conditions which corresponded to
measured relative humidities of 0.5±0.02 % and 65.0±0.2%, respectively.
Prior to every experiment involving ozonolysis, the system was prepared by thoroughly cleaning the
AFT and tubingand checking the stability of the gas flows and oxidant levels. Each experiment was
run back-to-back with and without the presence of ozone, with all other conditions remaining
identical, to allow measurement of both the initial OL particles before ozonolysis and the resulting
aerosol post ozonolysis.  Measurements were conducted once the aerosol concentration, size





distribution and ozone concentration had stabilised.  The system was flushed between runs to
prevent contamination from prior experiments.
**2.2      Scanning Mobility Particle Sizer (SMPS)**
To follow changes in the OL particle size distribution due to ozonolysis, an SMPS system
comprising an electrostatic classifier (TSI model 3080), neutralizer (TSI model 3076), long column
differential mobility analyzer (LDMA, TSI Model 3081) and condensation particle counter (CPC,
3022A) operating at 3.0 L/min and 0.3 L/min for sheath and aerosol flows, respectively, was used to
monitor the aerosol number and size distribution. A particle density equal to that of pure OL (0.891
g cm$^{-3}$) was applied to determine changes in the number, diameter, volume and mass distribution of
OL particles. Size distribution data were recorded and analysed using TSI AIM software v 8.1.
Aerosols were monitored in the size range of 15.1 – 667 nm, every 3 minutes, before and after
exposure to ozone. For each experimental setting, a minimum of 10 sizing scans were averaged.
**2.3      Aerosol Time of Flight Mass Spectrometer (ATOFMS)**
A detail description of the ATOFMS is given elsewhere (Gard et al., 1997; Sullivan and Prather,
2005; Dall'Osto et al., 2006); briefly, the ATOFMS is an on-line single particle instrument which
was applied here to measure the physical and chemical characteristics of OL aerosol before and
after ozonolysis. The ATOFMS deployed in this study (TSI, Model 3800-100) sampled particles in
the size range 100 – 3000 nm. For analysis of individual particle composition, time of flight mass
spectrometry (TOF-MS) was used with a laser (266 nm) desorption ionization (LDI) source.
The ATOMFS sampled from the AFT at a flow rate of 0.3 slm.  When an aerosol flow enters the
ATOFMS it is directed through an expansion nozzle and skimmers during which particles are
accelerated to a velocity characteristic of their aerodynamic size; the smaller the particle the higher
the speed. Velocities of individual particles, hence aerodynamic size, are measured via scattered



light from two CW timing lasers (532 nm) positioned a known distance apart.  The particle velocity
determines the timing of the subsequent LDI pulse to allow for interaction between particles and
LDI.  Once ionized, the molecular fragments from the particle are directed to both positive and
negative polarity time-of-flight mass spectrometers. Particles for which both size and mass spectra
are measured are referred to as hits, those particles for which size, but not a mass spectrum, are
measured are referred to as misses.
The detection of particles within the ATOFMS is reliant upon the LDI pulse generating ions,
requiring the particle to have sufficient absorption properties at 266 nm.  We found that the
interaction between the LDI and both OL and its ozonolysis products produced only negligible ions,
failing to meet the detection criteria of ATOFMS.  To increase the absorption properties of the
particles, the dye Nile Blue Sulphate (NBS) was introduced into the OL particles.  NBS effectively
absorbs light at the lasing wavelength of the LDI laser with absorption peaks occurring at 624, 325
and 276 nm) and therefore its introduction into the aerosol provides a methodology for efficient
ionization.  NBS was also separately introduced into calibration particles formed from each of the
primary products of OL ozonolysis (See Results section).
Although the ATOFMS response is matrix-sensitive and may not directly reflect the quantitative
composition of the measured aerosol (Gross et al., 2000; Fergenson et al., 2001; Bhave et al., 2002;
Sullivan and Prather, 2005; Allen et al., 2006; Dall'Osto et al., 2006), relative peak intensities can
be meaningful when particle constitution within a measured sample has not changed considerably
during a given investigation (Gard et al., 1998; Finlayson-Pitts and Pitts, 2000). In this sense, the
use of ATOFMS as a semi-quantifiable approach for the analysis of aged OL particles as a function
of particle size is justifiable because individual laboratory based particles studied under
experimentally controlled conditions are far less complex - and variable - than atmospheric aerosol
(Allen, 2004). Moreover, ATOFMS information on particle number and the abundance of different



species by particle size are possible with high time resolution (Dall'Osto and Harrison, 2012). Most
importantly, the relative signal intensities arising from the major oxidation products of the OL-O$_3$
heterogeneous reaction was studied using individual authenticated standards, the analysis of which
was used to calibrate signals arising from reacted oleic acid particles.
The ATOFMS particle sizing was calibrated with standardized polystyrene latex spheres (PSL,
calibrated diameters = 0.2, 0.3, 0.4, 0.56, 0.7, 1.3, 2, and 2.5 µm). Dilute PSL suspensions were
prepared in deionised water and nebulised using a constant output atomiser (TSI: Model 3076) at a
constant flow of 3.0 slm. The output of the nebulizer was sent through a diffusion dryer (TSI:
Model 3062) prior to entry into the ATOFMS sampling inlet to remove water. Analysis of the
measured velocity of the nebulised PSL particles as a function of their known particle sizes
provided a calibration curve for the ATOFMS particle sizing.
The mass scale of the ATOFMS was calibrated through nebulisation of two multi-element
commercial standards (see Chemicals and Reagents, below) and further standards prepared for this
study reflecting the principal products of the OL ozonolysis system : OL, AA, NN, NA and 4-
oxononanoic acid (4-ON) together with NBS (added chromophore). The obtained time of flight of
the resulting ions in each case was related to the corresponding expected m/z values. This approach
allowed for the ATOFMS response to be directly calibrated up to 563 Da.  m/z values reported here
beyond this limit have been calibrated using the molecular masses for those higher molecular
weight products for which identities have been proposed.  To assess the relative sensitivity of the
ATOFMS towards OL, NN, AA, NA and 4-ON, a standard mixture solution containing OL, NN,
AA, NA, and 4-ON at equimolar concentrations, plus traces of the laser dye, NBS, was prepared in
methanol and aerosolized using liquid atomisation.  The ATOFMS datasets were all processed
using MS Analyse and Microsoft Access software.



**2.4 Generation and Measurement of OL Aerosol and Ozone**
Ozone was generated by flowing 0.5 slm of synthetic air through a quartz photolysis tube
illuminated by a mercury UV pen-ray lamp (Ultra-Violet Products Ltd.). The ozone-containing air
(20 ppm at the ozone generator exit) then entered the AFT through an upstream inlet. The ozone
concentration was monitored downstream of the AFT using an ozone monitor (2B Technologies,
Model 205) with a sampling rate of 1.5 L/min. This elevated $O_3$ concentration allows use of short
contact times in the AFT and corresponds to an integrated exposure approximately equivalent to
about one day in ambient air [if potential within-particle diffusion limitations may be neglected].
Two approaches were used to generate OL aerosol. For the physical characterisation of OL aerosol
using the SMPS, liquid OL particles were generated by homogenous nucleation of pure OL vapour.
The low-volatile OL liquid was heated in a Pyrex vessel, in an insulated and temperature-controlled
oven (120±0.5 °C), to create OL vapour. To aid evaporation 0.5 slm of $N_2$ gas was bubbled through
the liquid OL. Subsequently, the vapour underwent homogeneous nucleation into OL aerosol
particles accelerated by the introduction of a 0.5 slm flow of $N_2$ gas at ambient temperature. The
resulting aerosol stream was passed through a reheating tube (200 °C) to homogenise the aerosol
and narrow the resulting particle size distribution. All steps in the procedure were kept at
temperatures significantly below the boiling point of OL (360 °C) to prevent pyrolysis. Overall,
these operating conditions generated OL aerosols with a geometric standard deviation $\sigma_g$ of 1.2, a
mass median diameter close to 400 nm and a total particle number concentration of $\sim 10^6$ cm$^{-3}$.
For the chemical product study of OL aerosol using the ATOFMS, a different method for
generating OL aerosol was required as a consequence of the necessity of doping the aerosol with
NBS dye (see discussion above). NBS was added at a low concentration ($5 \times 10^{-5}$ M) to liquid OL
and the mixture made up into 0.15 M solutions of OL in methanol. Subsequently, a constant output
atomiser (TSI, Model 3076) was used to generate a polydisperse aerosol ensemble from the





solution. The atomiser requires a constant flow rate of 3.0 slm for operation. Of this aerosol-
containing flow, 1.0 slm was directed into the AFT after it passed through a series of two silica gel
diffusion dryers (TSI: Model 3062) for the removal of methanol prior to entry to the AFT, while the
remaining flow was vented. Under the experimental conditions of this study, methanol is expected
to fully evaporate and indeed no methanol signal was detected in any of the ATOFMS mass spectra.
Generation of the other aerosols containing NBS (i.e. calibration standards) was performed in the
same manner.
**2.5     Chemicals and Reagents**
Liquid OL (purity > 99.0%), solid crystalline azelaic acid (AA, 98%), liquid nonanal (NN, 95%),
liquid nonanoic acid (NA, 96%) and liquid 4-oxononanoic acid (4ON, 95%) were purchased from
Sigma-Aldrich and used as supplied with no further purification steps attempted. For ATOFMS
mass calibration, multi-element standard solutions were used: standard A (900 µg/ml: Ba, Pb, Li, K,
Na & V in 5% $HNO_3$) and standard B (900 µg/ml: Ag and Mo in 5% $HNO_3$) obtained from TSI Inc.
The size calibration of the ATOFMS used polystyrene latex spheres (PSL, diameter range: 0.1 to
2.5 µm, Duke Scientific). The cationic laser dye, bis[5-amino-9-(diethylamino)benzo[a]phenoxazin-
7-ium] sulphate, Nile Blue Sulphate, used to modify the spectroscopic properties of OL particles
and laboratory grade methanol were obtained from Fisher Scientific. Synthetic air and nitrogen free
oxygen ($N_2$) (99.9% stated purity) were supplied by BOC and purified by passing through charcoal
traps (Grace Discovery Science, Efficiency<20 ppb) to remove trace hydrocarbon impurities.
Traces of water were removed from the gas lines using a trap (Sigma-Aldrich, Molecular Sieve 5A
Moisture Trap).



**3.     RESULTS**
**3.1      Effect of Ozonolysis on Particle Size**
The size distributions of ozonolysed and non-ozonolysed OL particles, under both dry and
humidified conditions, are shown in panels (a) and (b) in Figure 2. To aid comparison, the wet and
dry results are combined, and normalized to the peak signal of the non-processed particle, in Figure
2 panel (c).  It can be clearly observed that OL particles lose mass upon oxidation under both dry
and humidified conditions.  The modal diameter decreases from 400±14 to 346±13 nm and from
372±13 to 322 ±12 nm under the dry and humidified conditions, respectively, corresponding to a
total mass loss (estimated from the SMPS) of 32.5±5.3% and 26.3±2.5% respectively. Although the
observed mass loss upon oxidation is consistent with  previously reported qualitative and
quantitative observations (Morris et al., 2002; Katrib et al., 2004; Katrib et al., 2005b; Dennis-
Smither et al., 2012b; Lee et al., 2012), in view of the measured uncertainties the difference
between the two experiments, under different RH conditions, is statistically insignificant. The lack
of hygroscopic growth of the initial OL aerosol is consistent with the observations of Dennis-
Smither et al. (2012a) and Hung et al. (2005).
The decrease in the mobility diameter of OL particles upon oxidation is likely due to the formation
of volatile products that partition from the particle to the gas phase.  This is consistent with previous
studies that have identified gas phase products.  In particular the highly volatile nonanal (NN) and
the semi-volatile nonanoic acid (NA) products have been previously identified (Moise and Rudich,
2002; Smith et al., 2002; Thornberry and Abbatt, 2004; Vesna  et al., 2009; Lee et al., 2012). The
formation of low volatility products, which remain in the particle phase, and typically possess
greater densities ($\rho$) than OL ($\rho = 0.891$ g cm$^{-3}$), such as azelaic acid (AA, $\rho = 1.225$ g cm$^{-3}$) and 9-
oxononanoic acid (ON, $\rho = 1.019$ g cm$^{-3}$) may also have an influence.



The fact that less mass loss was observed under humidified conditions than under dry conditions
(although the difference was statistically insignificant), suggests that the presence of water might
have an effect on the oxidation mechanism. Such a conclusion has been reported in the literature for
similar oxidation conditions (Gallimore et al., 2011). It has been suggested that the presence of
particle phase water leads to different CI reactivity (i.e. an alternate CI fate) which results in the
formation of less volatile products (such as organic acid formation) and hence less particle mass
loss (Gallimore et al., 2011). This is surprising because OL and its oxidation products, even though
they are smaller and contain more hydrophilic chemical moieties (carboxylic acid and hydroxyl
groups), have low solubility in water. Results from other experimental studies indicate that OL has
mild hygroscopicity (Andrews and Larson, 1993; Kumar et al., 2003; Vesna et al., 2008)
Furthermore thermodynamic modelling indicates that OL and ozonolysed OL particles show small
but non-negligible hygroscopicity at RH 65% (Lee et al., 2012). It should be noted that
hygroscopicity is defined by mass or diameter growth upon uptake of water referenced to the dry
state. Since the molar mass and molar volume of water is much smaller than OL and its oxidation
products, the molar ratio of water to OL or its oxidation products is much greater than the reported
hygroscopicity would suggest. Dennis-Smither et al. (2012a,b) report a reduction in particle size
with oxidation of OL and that hygroscopicity increases with oxidative aging. While Lee et al.
(2012) observed no change to the reaction scheme of OL with increased humidity, Vesna et al.
(2009) report changes in product yields.
**3.2    Chemical Characterisation of OL and Ozonolysed OL Aerosol**
Particles leaving the AFT were sampled by the ATOFMS. Ionisation within the instrument leads to
ions which enter two time-of-flight mass spectrometers. Both positive and negative mass spectra
were recorded for all detected particles. The negative mass spectra show a greater abundance of
peaks which is expected based upon the predicted reaction pathways (see Scheme 1) because most
OL oxidation products contain carboxylic acid and/or aldehyde moieties which are principally



detected as deprotonated molecular ions [M-H]⁻ formed via proton abstraction. To allow easier
interpretation and minimise fragmentation, the LDI laser fluence of the ATOFMS was kept very
low (0.4–0.8 mJ per pulse) as compared to other studies (1.3–1.6 mJ) (Silva and Prather, 2000;
Dall'Osto and Harrison, 2012). Nevertheless, it should be noted that some fragmentation of ions in
the ATOFMS system is expected and this can lead to difficulty in interpretation.
**3.3     Mass Spectrometric Analysis of NBS and OL**
To probe for any influence of NBS on the chemistry of OL ozonolysis two preliminary tests were
carried out.   Firstly the mass spectra of NBS aerosol with and without $O_3$ present were recorded
(Figure S1a).  Secondly the mass spectrum of aerosol composed of NBS and OL without $O_3$ was
recorded (Figure S1b).  NBS displays three strong signals in the positive spectrum at m/z +274 and
+308 assigned to the cationic dye fragments and +318 assigned for the non-fragmented dye cation.
The two strong signals in the negative spectra at m/z -96 and -80 correspond, respectively, to the
sulphate ($SO_4^{2-}$) and sulphite ($SO_3^{2-}$) ions of the dye. No changes in the NBS spectra were detected
when NBS particles were exposed to ozone (Figure S1a).  When OL was added to NBS solution in
the absence of ozone, the only additional peaks observed were the deprotonated mass signals from
OL and the OL dimer. It should be noted that NBS peaks were always observed in all mass spectra
shown henceforth. However, for the ease of the analysis, all NBS peaks are omitted from the
presented mass spectra.
**3.4     Mass Spectrometric Analysis of OL Oxidation Products**
In order to quantify the products of OL ozonolysis the relationship between the mass spectral signal
and the concentration needs to be quantified.  Standards for OL, AA, NN and NA were all
commercially available. However, ON was not commercially available but its isomer 4-
oxononanoic acid (4-ON) was.  The detection of ON via the ATOFMS is expected to be near-
identical to that of 4-ON.  ON contains a carboxylic acid and an aldehyde group whereas 4-ON





contains a carboxylic acid and a ketone group. Both acids, from ON and 4-ON, are expected to
exhibit a deprotonated carboxylate ion [M-H]$^-$ and were detected in the negative ion mass spectrum
at m/z -171.  Standards for the high molecular weight oxidation products are not available
commercially and are difficult to synthesise and hence were unavailable for this study.
Figure 3 shows the positive and negative mass spectra of the individual standards. Unreacted OL
aerosol (282 Da) was seen as a deprotonated OL molecular ion at m/z = -281 and as a singly
deprotonated dimer at m/z -563. The deprotonated molecular ions of AA (MW = 188 Da), NA (158
Da), NN (142 Da) and 4-ON (172 Da) are seen as major peaks at m/z -187, -157, -141 and -171
respectively, while the singly deprotonated dimers of AA and NA were measured at m/z -375 and -
315, respectively. In the positive ion mass spectra of AA a signal at m/z +155 was attributed to the
fraction of the molecule after the loss of an HO$_2$ fragment [M-HO$_2$]$^+$. The molecular ion peak
measured at m/z +113 in the NA positive ion mass spectra was assigned to the molecular fragment
of NA after the loss of CO$_2$H, [M-CO$_2$H]$^+$. It is important to note that the ATOFMS intensity for
individual ions is dependent upon the total composition of the particle.
To investigate ATOFMS sensitivity towards OL and the primary ozonolysis products, a solution
containing an equimolar mixture of OL, AA, NA, NN and 4-ON was examined under the same
experimental conditions as the oxidation experiments. Figure 4 shows the results of the ATOFMS
(relative) sensitivity to the deprotonated molecular ions of the five component mixture.  An average
spectrum comprised of 200 spectra was used for the analysis.  It is evident that the ATOFMS
sensitivity varies considerably between the different compounds.
The figure indicates that the signals observed for the five component mixture is in the following
order: AA >> ON > OL > NA > NN.  The much greater signal from AA compared to the other
species can be partially understood because it is a dicarboxylic acid and thus has a greater




possibility of being ionized to the carboxylate ion compared to the species containing only one
carboxylic acid moiety.  ON and OL have similar signals which is expected since they both contain
one carboxylic acid group.  The signal from NN is the smallest probably because it only contains an
aldehyde moiety which is more difficult to ionize compared to carboxylic acids. Previous studies on
NN detection using soft ionisation and ultra-high resolution mass spectrometry also showed weak
NN peaks in the mass spectra. The authors attribute this observation partly to the poor signal
produced from aldehydes by the deployed techniques (Grimm et al., 2006; Hosny et al., 2013).  The
high volatility of NN will also lead to the partitioning from the aerosol to gas phase thereby
reducing the particle phase concentration, although it is noted that this process will also happen in
the processed oleic acid particles.
The relative peak areas of the five component mixture were converted into sensitivity correction
factors (SCF) that allow for relative concentrations of these components to be estimated within the
ozonolysed OL particles.   It should be noted that this approach – as in many other analytical
approaches – is not sensitive to any matrix effect of ozonolysed OL particles which might have an
impact of suppressing or enhancing ATOFMS signals of individual components. To minimize
possible matrix complications, the relative peak intensities of the individual components within
each measured sample were used to describe the component distribution of the chemically aged OL
polydisperse aerosol. As mentioned in Section 2.3, satisfactory ATOFMS measurement of the aged
OL particles required improving the efficiency of the LDI by optically modifying the matrix of OL
particles using added NBS laser dye. This modification resulted in a substantial enhancement in
ATOFMS detection of particles apparently by maximising the absorption efficiency of the pulsed
laser by the particles.





**3.5    Mass Spectrometric Analysis of Aged OL Particles**
The size binned ATOFMS data show distinct differences in composition between small particles
($D_P$<0.3 µm) and large particles ($D_P$>0.3 µm).  Small particles are characterised by the presence of
compounds with molecular weights lower than the parent OL molecules (m/z<282), whereas large
particles are characterised by significant formation of higher molecular weight (HMW) compounds
(m/z>282).  Smith et al. (2002) measured ozone uptake by OL particles as a function of particle
size.  They found that the effective uptake coefficient of ozone decreased with increasing particle
size, concluding that this resulted from the reaction being limited by the diffusion of OL within the
particle.  Under such conditions, reactions between intermediate products and oleic acid become
important, hence leading to the formation of higher molecular weight species (Hung et al., 2005;
2007; Katrib et al., 2004).  Reactions between ozonolysis products and Criegee intermediates also
lead to high molecular weight products when the supply of ozone is limited (Reynolds et al., 2006;
Zahardis et al., 2006).  Our interpretation of the mass spectral information gained from the
ATOFMS appears in Table 1.
**3.6    Mechanisms and Mass Spectrometric Analysis of Small Particles**
The mass spectrum of small oxidised OL particles (average of 300 spectra) is shown in Figure 5.
Three of the primary ozonolysis products: AA, ON, and NA, were observed at m/z -187, -171 and -
157, respectively. Peaks measured at m/z=-143, -329 and -439 could be indicative of octanoic acid
(OcA) (144 Da), the primary ozonide (POZ), (see Scheme 1) and the secondary reaction product of
CI2 with OL (reaction R 1), respectively. The unidentified mass spectrum peaks are likely due to
fragments (daughter ion peaks) or aggregates of ions.  No useful insights were gained from the
positive mass spectrum.



**R 1**
**3.7**      **Mechanisms and Mass Spectrometric Analysis of Large Size Particles**
The three frames, a, b, and c, in Figure 6 show negative and positive ion mass spectra of large ($D_p >$
0.3 μm) aged OL particles (average of 700 spectra). The negative ion mass spectrum is more
complex but compatible with the hypothesis of secondary chemistry and the formation of HMW
products. Generally, the spectrum demonstrates a strong peak of unreacted parent OL at m/z -281
and the appearance of the major oxidation products of ozone exposure. All four primary products
(AA, ON, NA and NN) are found to be in the particle phase including the highly volatile NN which
was not detected in the small particles. The presence of NN in large particles is consistent with the
observations of Dennis-Smithers et al. who studied supermicron sized particles (Dennis-Smither et
al., 2012b). The signals of AA and OcA which form through the molecular rearrangement of CI1,
were not detected in the larger particles, which might reveal that isomerization is predominant in
smaller particles and that it is the only formation source for these products. Both negative and
positive average mass spectra are characterized by the presence of HMW products presumably
formed by secondary association reactions of primary reaction products.
Previous studies have reported evidence of secondary chemistry occurring within the ozonolysis of
OL particles (Katrib et al., 2004; Hearn et al., 2005; Ziemann, 2005; Reynolds et al., 2006; Zahardis
et al., 2006; Zahardis and Petrucci, 2007; Lee et al., 2012; Hosny et al., 2013). The presence of the
liquid condensed phase substrate for the CIs minimizes their molecular rearrangement (Katrib et al.,
2004) via a solvent cage effect (Park et al., 2006) and maximizes their lifetimes. Therefore, reaction
probability of CIs with their corresponding carbonyl compounds to form secondary ozonide (SOZ)
or with the alkene functionality becomes more significant (Neeb et al., 1998; Moise and Rudich,
2002; Zahardis et al., 2005).





Table 1 proposes molecular structures for the observed HMW mass spectral signals based upon
polymerization mechanisms previously proposed in the literature (Smith et al., 2002; Hearn and
Smith, 2004; Katrib et al., 2004; Hung et al., 2005; Zahardis et al., 2005; Reynolds et al., 2006;
Zahardis et al., 2006; Hung and Ariya, 2007; Zahardis and Petrucci, 2007). For instance, the peak at
m/z -297 can be assigned for two isomeric compounds, 9-oxooctadecanoic acid (Ox1) or 10-
oxooctadecanoic acid (Ox2), depending on the type of the CI formed and the geometry of the CI
addition across the double bond of OL to form the $C_{27}$ peroxide which can cleave to yield the
primary products ON or NN. Similarly, the detected signal at m/z -327 is determined as a reaction
product between the OL double bond and CI1 to form 9-oxooctadecanedioic acid (Ox3). Scheme 2
illustrates a proposed mechanism and product structures formed as a result of the reaction between
CI1 and the alkene functionality of OL.
The presence of peaks correspond to Ox1 or Ox2 in the OL-$O_3$ heterogeneous reaction has been
reported previously (Hearn and Smith, 2004; Zahardis et al., 2006; Hung and Ariya, 2007) while the
reaction mechanism and product structures were first hypothesized by Katrib and co-workers
(Katrib et al., 2004).
**Scheme 2:** Potential mechanism and product structures formed as a result of the reaction between
CI1 and the alkene functionality of OL.



The existence of the carboxylic acid functionality in the oxidation products offers reaction sites for
other CIs. The measured negative ion peaks at m/z -1019, -1189, and -1207 provide more evidence
of the incorporation of Ox1 or Ox2 as linear polymerization propagators. However, the ion peak
measured at m/z -644 is consistent with the formation of the ozonolysis product resulted from the
reaction of two ions of CI1 and the two carboxylic moiety of one molecule of Ox.
The addition of CI2 terminates the polymerization reaction as the –CH$_3$ group cannot react further
(Hung et al., 2005). The major HMW products ions at m/z -1025, -1079, -1214, -1292, -1310, -
1346, -1438, and -1524 may also correspond to polymerization products of Ox3 joined with other
propagators. Products related to the secondary reaction of ON are also observed in a number of
peaks. The negative ion at m/z -341 is most likely due to the combination of ON and CI1 to form
AAHP with a loss of one molecule of water. Although the same combination can lead to the
formation of SOZ, however, in such arrangement a water molecule cannot be removed. The in situ
dehydration of AAHP was first observed and described by Zahardis and co-workers (Zahardis et al.,
2006). The resultant proposed structure suggests possible additional moieties linking from molecule
ends, the aldehyde and carboxylic acid groups, thereby growing into a HMW linear polymer;
Scheme 3 explains.
**Scheme 3.** Suggested reaction pathways and products of the secondary reaction between CI1 and
ON.



**3.8     Particle Composition as a Function of Particle Size**
The ATOFMS technique allows for particle composition data to be collected as a function of
particle size, thereby permitting the size dependence of the chemical aging of aerosols to be
determined.  This section investigates how the composition of aged OL particles changes, under dry
and humidified conditions, as a function of particle size.  The analysis only investigates the
following compounds which have clear MS peaks:  OL, NN, AA, ON, NA and HMW compounds
(m/z > 282).  The conversion from measured peak area to molar concentration is achieved using the
standard calibration factor (SCF). Since an accurate SCF for the HMW compounds is difficult to
determine due to the lack of standard laboratory calibrants, an upper limit value for the HMW SCF
was estimated using a mass balance approach.  In particular, the ratio between the OL signal to the
total peak signals of the four primary oxidation products (AA, NN, NA and ON) was used to
estimate the ratio between unreacted OL and the products. The obtained ratio of each product was
corrected using the SCF to estimate the absolute molar ratio and hence the composition of the
primary products in the particles. The reduction in the molar ratio between the four products and
unreacted OL was used to estimate the SCF for the HMW (1 mole of OL produces 2 moles of the
primary products (Scheme 1)).
**3.9     Composition of Aged OL Particles Under Dry Conditions**
Figure 7 provides the compositional analysis, by applying SCFs to the raw ATOFMS data, of 1000
aged OL particles that were ozonolysed under dry conditions.  The size dependent composition is
given in Figure 7a and the average composition of the aged OL polydisperse aerosol is shown in
Figure 7b.  The average composition was achieved by taking the mean of all mass fractions
analysed regardless of size; since most particles were in the size range 0.3-0.5 µm this analysis will
be biased towards this size fraction.  The overall size distribution of the analysed particles was in
the range 0.2-3 µm as measured by ATOFMS, see Figure 7c.



It can be clearly observed that the composition of aged particles is highly dependent on particle
size. Firstly, larger particles contain more unreacted OL than smaller particles. This can be
understood in terms of OL diffusivity within the particle and ozone flux into the particle. As the
reaction progresses and oxidation products evolve at the reacted surface layer, both OL diffusivity
within the particle and ozone diffusion to the bulk are reduced. Smith et al. (2002) reported a low
decay of OL in larger particles which was attributed by the limited OL diffusivity within the large
particles. However, OL molecules in smaller particles (at a diameter size of 200 nm and less) are
homogeneous in concentration and thus particles at this size and below are well mixed (Smith et al.,
2002; Pfrang et al., 2010) facilitating the oxidation process and therefore the smaller the portion of
unreacted OL. Additionally, in a simulation study, Shiraiwa et al. (2010) showed that ozone uptake
drops within the first second of the heterogeneous reaction. The authors explained the drop in the
calculated ozone uptake on the basis of the rapid decrease in OL concentration at the particle
surface while it remains constant in the inner particle bulk. It is therefore reasonable to conclude
that the larger the particle the less the molar ratio of $O_3$:OL in the particle phase and hence the
smaller the degree of oxidation and the greater proportion of unreacted OL. The fraction of NN in
the particle tends to increase with particle size. The smallest size fraction contained only negligible
amounts of NN. The vapour pressure of NN is 49.3 Pa at 25ºC (Daubert and Danner, 1989) and
hence it is a highly volatile species which tends to partition to the vapour phase. In previous studies
NN has been identified as one of the major volatile products (Moise and Rudich, 2002; Thornberry
and Abbatt, 2004; Zahardis and Petrucci, 2007; Vesna et al., 2009; Dennis-Smither et al., 2012b),
while few investigators detected NN in the particle phase using supermicron (King et al., 2004) and
supported millimetre sized particles (Hung and Ariya, 2007) . It is likely that NN has a low
condensed phase yield in small particles due to its volatility that results in fast partition to the
vapour phase. NN signals, however, were frequently enhanced in large particle spectra since its
concentration in these particles was evolved from the secondary reaction within the particles, and
products in these particles might reduce the effective vapour pressure of NN. Loss as vapour would



also require diffusion within the condensed phase, which takes longer to reach the surface in the
larger particles.
The mole fraction of AA, which arises from the rearrangement of CI1, was significantly larger in
small particles compared to larger particles. It diminished appreciably in the most reacted larger
particles suggesting the involvement of AA in the secondary chemistry associated more with larger
particles.  NA which arises from CI2 did not show a similar trend to AA, and was detected in small
and comparable fractions in most size bins. This may reflect the volatile nature of NA which would
reduce its impact in secondary chemistry as it tends to partition away from the particle. The steady
decrease in ON mole fraction as particle size increases possibly indicates the involvement of ON in
the secondary reactions associated with larger particles.
Figure 7a suggests that reactions in larger particles can appreciably enhance the formation of HMW
($> 282$ m/z ) products.  The chemical composition of larger particles, compared to smaller particles,
is characterised by more unreacted OL, increased HMW products and less AA.  The low
concentration of OL in small particles, however, minimizes these secondary pathways and
maximizes the formation of the SOZ which subsequently dissociates to form the four primary
oxidation products. The quantification analysis of processed OL particles by Katrib and co-workers
showed no trend for AA with increasing OL layer thickness, and high ON yield as compared to AA
and NA yields was observed (Katrib et al., 2004), very consistent with our findings. They attributed
their finding to the reactivity of the AA and NA precursors, i.e. CIs, being scavenged by OL.  The
presence of HMW products likely leads to greater aerosol viscosity and hence lower diffusion rates
within the particle (Hosny et al., 2016). Hence larger particles will likely have slower diffusion rates
which may in part explain the greater proportion of NN in the larger particles.  Moreover, measured
viscosity of chemically aged aerosol might therefore be expected to vary substantially with -





increase with - particle size, of relevance to the interpretation of aerosol particle viscosity
measurements performed on individual super-micron samples.
**3.10    Composition of Aged OL Particles under Wet Conditions**
Figures 8 provides the corresponding compositional analysis, by applying SCFs to the raw
ATOFMS data, of 1000 aged OL particles that were ozonolysed under humidified conditions.
Significant differences in both the size dependent and average distribution of components between
the dry and humidified oxidation runs were observed.
The greatest differences between the dry and wet oxidation data, averaged over all size bins for the
complete polydisperse aerosol ensemble, is the reduction in HMW products from 18.8% under dry
conditions compared to 4.1% under humidified conditions. The proportion of unreacted OL
increases under humidified conditions (31.7%) compared to dry conditions (13.7%). Whilst OL is
negligibly hygroscopic, ozonolysed OL is slightly hygroscopic (Lee et al., 2012). This slight
hygroscopicity suggests that particle phase water could act as a reactant.  The differences in
composition between the dry and the humidified experiments could be explained by the preferential
reactivity of CIs with water molecules compared to OL thereby resulting in a lower OL
consumption under wet conditions.   This chemistry results in less destruction of additional OL
molecules via the secondary pathways, and hence the higher portion of unreacted OL and the
limited amount of HMW products which are observed concomitantly under wetter conditions. This
also may explain the observed decline in the NN signal from 4.3% in dry oxidation to 0.7% in wet
oxidation. The secondary chemistry associated with the formation of NN in the bulk of the droplet
is suppressed by the reaction of CIs with water. Lower particle viscosity within the particles
processed under humid conditions, due to the reduced HMW fraction, may also contribute to the
lower NN concentration compared to that found under dry conditions.  The size distribution data
from the SMPS, which indicated less mass loss during the humidified oxidation, supports this





argument. However, small particle sizes show a different trend in these experiments. The observed
loss of OL in the smallest particles (300 nm) increased slightly under humid condition (Figure 8 and
7). This result is partially supported by the study of Vesna et al. (2009), who used smaller particles
(geometric mean diameter of 78 nm), and reported a similar trend with the smallest size bin of the
particles in this study. The small increase in the loss of OL and the increase in the product yields of
ON observed in response to the increase in the RH are consistent with our findings, although Vesna
et al. (2009) report a higher abundance of "unidentified products".
**3.11      Implications for Ageing Atmospheric Organic Aerosol**
The observed reactions could have consequences for the ability of OL-derived particles (and OA of
comparable functionality) to act as cloud condensation nuclei (CCN).  Hygroscopicity greatly
enhances the ability of particles to act as CCN, and consequently, oxidised particles containing a
larger proportion of shorter-chain polar molecules are likely to be the most effective CCN for a
given particle size.  King et al. (2009) use Köhler theory to demonstrate that the more oxidised
particles will activate at a lower supersaturation than unreacted OL particles.  However, the
reduction in particle size which accompanies the loss of NN to the vapour phase will work in the
opposite sense of increasing the critical supersaturation due to an increased Kelvin effect, at least
for the smaller particles.  However, since particles in the atmosphere are typically mixed and hence
unlikely to comprise purely OL, even when emitted, such a discussion is likely to be of very limited
relevance to atmospheric behaviour.
Heterogeneous reactions of organic particles directly alter the size, density and chemical
composition of the particles. These are the key parameters controlling the particle's lifetime in the
atmosphere and optical properties. While the size of the particle has considerable impact on the
deposition velocity, wet scavenging efficiency and scattering of light, the identity of the species
within the particle is the principal characteristic driving light absorption. The aging of organic



aerosol can result in the formation of light absorbing species. The large numbers of organic
functionalities in SOA such as carboxylate, hydroxyl, ketone and aldehyde groups may result in an
absorbing matrix that can exhibit optical properties dramatically different from those of parent
molecules. Particles containing light absorbers may lead to heating of the lower atmosphere
resulting in positive global radiative forcing. The impact of SOA on global radiative forcing is thus
one of the largest uncertainties in atmospheric science. On the other hand, the presence of absorbing
compounds in organic particles can stimulate photosensitization processes which might lead to
either reduction and oxidation of intermediates and products (Kolb et al., 2010). For the reasons
outlined above, because of the internally mixed nature of airborne particles, closer examination of
the properties of oxidised OL particles is unlikely to be of major relevance to prediction of the
properties of atmospheric aerosol.
**4.      CONCLUSION**
This study demonstrates a link between the particle size and reaction mechanisms within OL
aerosol. Aged finer particles are likely to be more hydrophilic due to the oxidation of the OL in the
particles and the formation of an early generation of more polar, and hence more hygroscopic
oxidation products. However, the combination of unreacted OL, HMW products and volatile
product observed in large particles suggests overall hydrophobicity of larger particles. The
difference in the reactivity of OL at different relative humidities with less OL destruction observed
under humid conditions can be explained by the preference of CIs for reaction with water molecules
over the reaction with OL, which results in less oligomerisation.
There has been some speculation in the literature as to the effects of oxidation upon atmospherically
relevant properties of OL particles, such as their ability to act as CCN. However, the relevance of
extrapolation to the atmospheric context, except in a generic sense, is extremely limited due to the
complex internal mixing of atmospheric particles, even when emitted, and hence the extreme



improbability of pure OL particles existing in the atmosphere. However, in our view, the value of
studies such as this is in the enhanced mechanistic understanding gained from treating OL oxidation
by ozone as a model system. In particular, there is limited understanding of processes leading to the
formation of highly functionalised oxidised high molecular weight products, such as those observed
in this work.
**ACKNOWLEDGEMENTS**
S. Al-Kindi is pleased to acknowledge financial support for her studentship from the Government of
Oman. This work was funded in part by the Natural Environment Research Council (NERC)
project NE/G009031/1, Artificial Chemical Ageing of Ambient Atmospheric Aerosol. Original
research data are available from the authors on request.



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





**TABLE LEGENDS**
**Table 1:** A summary of proposed components and possible propagator combinations contributing to of
observed mass spectral peaks corresponding to 44 oxidation products of the OL-$O_3$ system.
**FIGURE LEGENDS**
**Figure 1:** The experimental setup for the study of the heterogeneous oxidation of OL aerosol.
**Figure 2:** Particle mass size distributions for (a) pure and oxidised OL aerosol under dry
condition (RH $0.5 \pm 0.02\%$), (b) pure and oxidised OL aerosol under wet condition
(RH $65.0 \pm 0.2\%$) and (c) normalised particle size distribution of pure and oxidised
OL aerosol under dry and wet conditions. Each curve represents the mean average of
10 measurements with accompanying standard deviation ($\sigma$).
**Figure 3:** Positive and negative ion mass spectra of OL, AA, NA, NN and 4-ON. Each spectrum
presented represents the average from 100 mass spectra.
**Figure 4:** Relative peak area signals from aerosol particles generated from an eqimolar mixture
of OL, AA, NA, NN and 4-ON. Peak areas are an average from 200 mass spectra.
**Figure 5:** Averaged ATOFMS negative and positive ion mass spectra of small processed OL
particles (Dp<0.3 µm).
**Figure 6:** Average mass spectra of dry aged OL particles (Dp>0.3 µm): (a) negative ion MS (b)
zoom in plot of figure (a) and (c) positive ion MS.
**Figure 7:** ATOFMS data analysis of: (a) mole fraction of ozonolysed OL particles, under dry
conditions, as a function of particle size, (b) total particle composition as given by
mole fraction for all size fractions, and (c) Corresponding size distribution of the aged
OL aerosol measured by ATOFMS.
**Figure 8:** ATOFMS data analysis of: (a) mole fraction of ozonolysed OL particles, under
humidified conditions, as a function of particle size, (b) total particle composition as
given by mole fraction for all size fractions, and (c) corresponding size distribution of
the aged OL aerosol measured by ATOFMS.





1 **Table 1:** A summary of proposed components and possible propagator combinations contributing to of
2 observed mass spectral peaks corresponding to 44 oxidation products of the OL-$O_3$ system

| MW | MS signature | | No of composed components | | | | | | | | Dehydration |
| | - m/z | + m/z | NN | OcA | AA or CI1 | ON | NA or CI2 | Ox1 or Ox2 | Ox3 | OL | (-n$H_2O$) |
|---|---|---|---|---|---|---|---|---|---|---|---|
| 142 | 141 | | 1 | | | | | | | | |
| 144 | 143 | | | 1 | | | | | | | |
| 158 | 157 | | | | | | 1 | | | | |
| 170 | 169 | | | | 1 | | | | | | 1 |
| 172 | 171 | | | | | 1 | | | | | |
| 188 | 187 | | | | 1 | | | | | | |
| 298 | 297 | | | | | | | 1 | | | |
| 314 | 313 | | | 1 | 1 | | | | | | |
| 328 | 327 | | | | | | | 1 | | | |
| 342 | 341 | | | | 1 | 1 | | | | | |
| 422 | 421 | 377[*] | | | | | 1 | | | 1 | 1 |
| 440 | 439 | 423[†] | | | | | 1 | | | 1 | 0 |
| 528 | 527 | 466[*†] 483[*] 547[‡] | | | 3 | | | | | | 3 |
| 644 | 643 | 599[*] | | | | | 2 | | 1 | | 0 |
| 656 | 655 | | | | 2 | | 2 | | | | 2 |
| 768 | 767 | | | | 1 | | 2 | | | 1 | 1 |
| 786 | 785 | 753[‡] | | | 1 | | 2 | | | 1 | 0 |
| 810 | 809 | 793[†] 777[‡] | | | 2 | 1 | 2 | | | | 3 |
| 844 | 843 | 811[‡] | | | 3 | | 2 | | | | 2 |
| 864 | 863 | 819[*] 831[‡] | | | 2 | 1 | 2 | | | | 0 |
| 880 | 879 | | | | 3 | | 2 | | | | 0 |
| 894 | 893 | 861[‡] | | | 3 | 1 | 1 | | | | 0 |
| 950 | 949 | 917[‡] | | | 3 | | 1 | | | 1 | 3 |
| 968 | 967 | 935[‡] | | | 3 | | 1 | | | 1 | 2 |
| 974 | 973 | | | | 2 | | 2 | | | 1 | 0 |
| 986 | 985 | 953[‡] | | | 3 | | 1 | | | 1 | 1 |
| 1002 | 1001 | | | | 3 | | 3 | | | | 2 |



| | | | | | | | | | | | | |
|---|---|---|---|---|---|---|---|---|---|---|---|---|
| 1020 | 1019 | 1003† | | | 3 | | 1 | 1 | | | | 0 |
| 1026 | 1025 | | | | 4 | | | | 1 | | | 3 |
| 1038 | 1037 | 1005⸓ | | | 3 | | 3 | | | | | 0 |
| 1051 | 1050 | 1070‡ | | | 4 | | 2 | | | | | 1 |
| 1068 | 1067 | 1023*  1106‡ | | | 4 | | 2 | | | | | 0 |
| 1080 | 1079 | | | | 4 | | | | 1 | | | 0 |
| 1162 | 1161 | 1112†⸓ | | | 3 | | 2 | | | | 1 | 0 |
| 1178 | 1177 | | | | 3 | | 4 | | | | | 1 |
| 1190 | 1189 | | | | 4 | | 1 | 1 | | | | 1 |
| 1196 | 1195 | | | | 3 | | 4 | | | | | 0 |
| 1208 | 1207 | | | | 4 | | 1 | 1 | | | | 0 |
| 1214 | 1213 | | | | 5 | | | | 1 | | | 3 |
| 1292 | 1291 | | | | 3 | 1 | | | 1 | | 1 | 3 |
| 1310 | 1309 | | | | 3 | 1 | | | 1 | | 1 | 2 |
| 1346 | 1345 | | | | 3 | 1 | | | 1 | | 1 | 0 |
| 1438 | 1437 | | | | 6 | | | | 1 | | | 1 |
| 1524 | | 1458⸓ | | | 3 | | 4 | | 1 | | | 0 |

1    *M-CO$_2$H, ⸓M-HO$_2$, †M-OH and ‡M+H$_3$O.





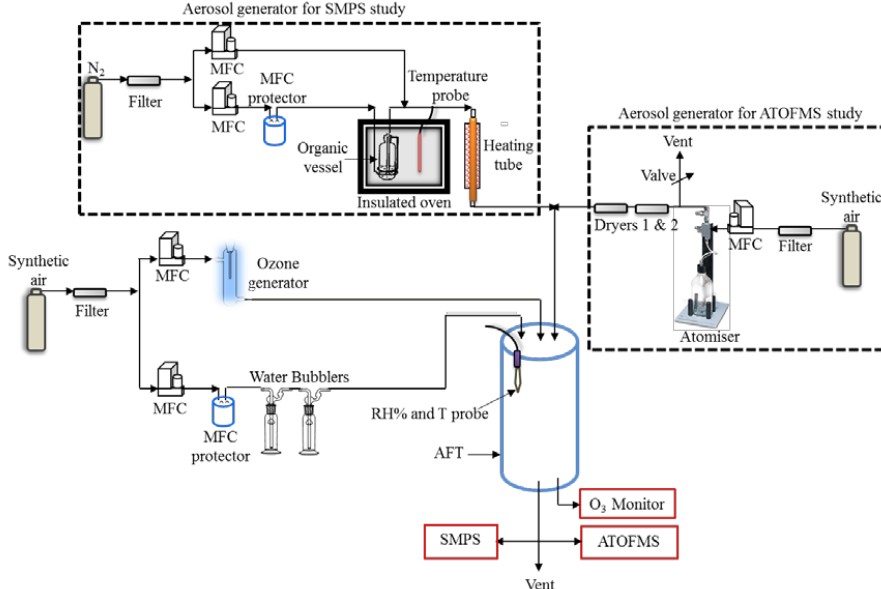

2      **Figure 1:** The experimental setup for the study of the heterogeneous oxidation of OL aerosol



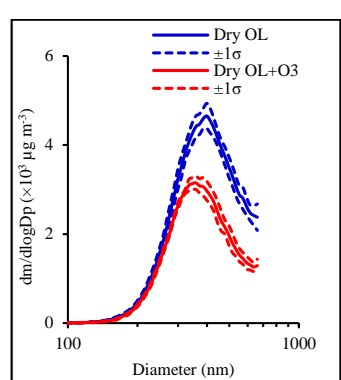 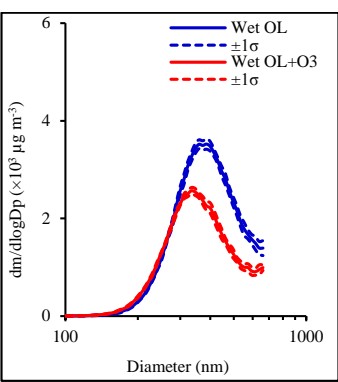 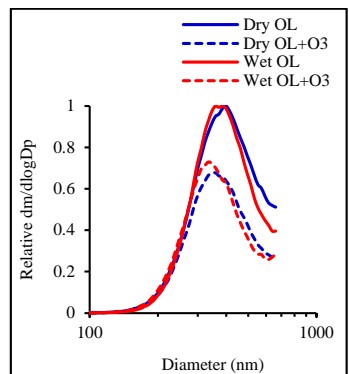

**Figure 2:** Particle mass size distributions for (a) pure and oxidised OL aerosol under dry condition (RH 0.5 ± 0.02%), (b) pure and oxidised OL aerosol under wet condition (RH 65.0 ± 0.2%) and (c) normalised particle size distribution of pure and oxidised OL aerosol under dry and wet conditions. Each curve represents the mean average of 10 measurements with accompanying standard deviation (σ)





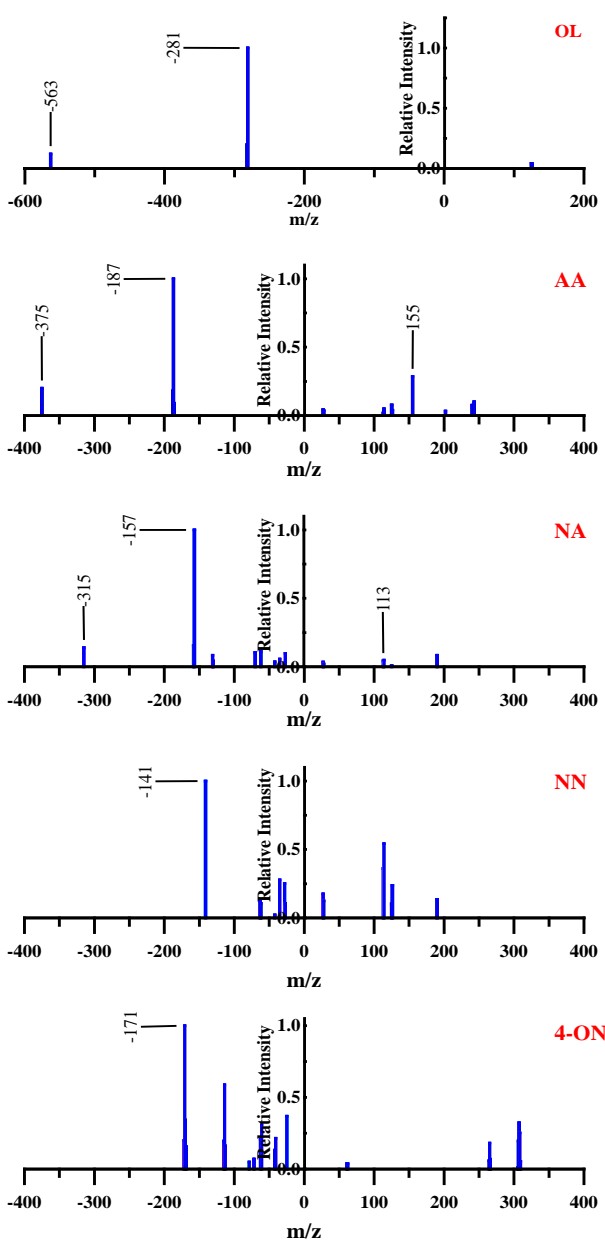

2  **Figure 3:** Positive and negative ion mass spectra of OL, AA, NA, NN and 4-ON. Each spectrum
3  presented represents the average from 100 mass spectra



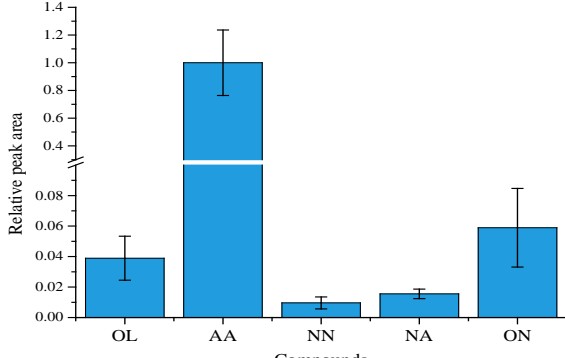

2 **Figure 4:** Relative peak area signals from aerosol particles generated from an eqimolar mixture of
3 OL, AA, NA, NN and 4-ON. Peak areas are an average from 200 mass spectra
4





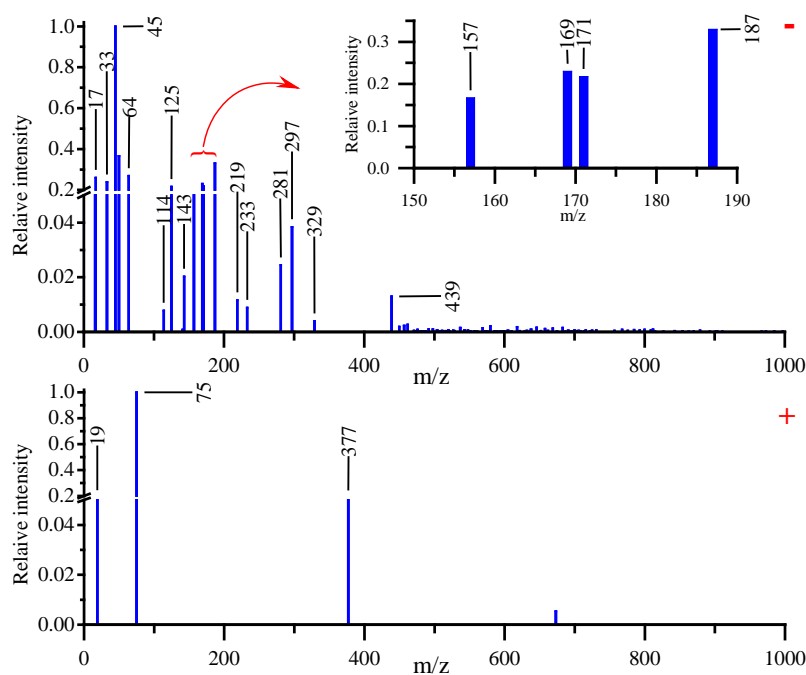

2  **Figure 5:** Averaged ATOFMS negative and positive ion mass spectra of small processed OL

3  particles (Dp<0.3 μm)



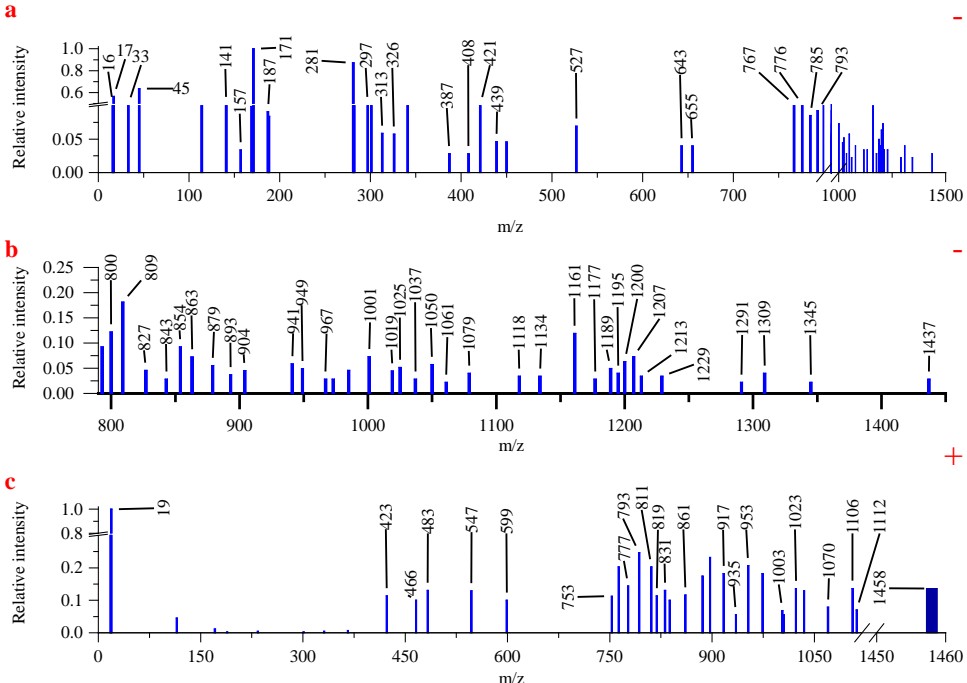

2 **Figure 6:** Average mass spectra of dry aged OL particles (Dp>0.3 µm): (a) negative ion MS (b)
3 zoom in plot of figure (a) and (c) positive ion MS



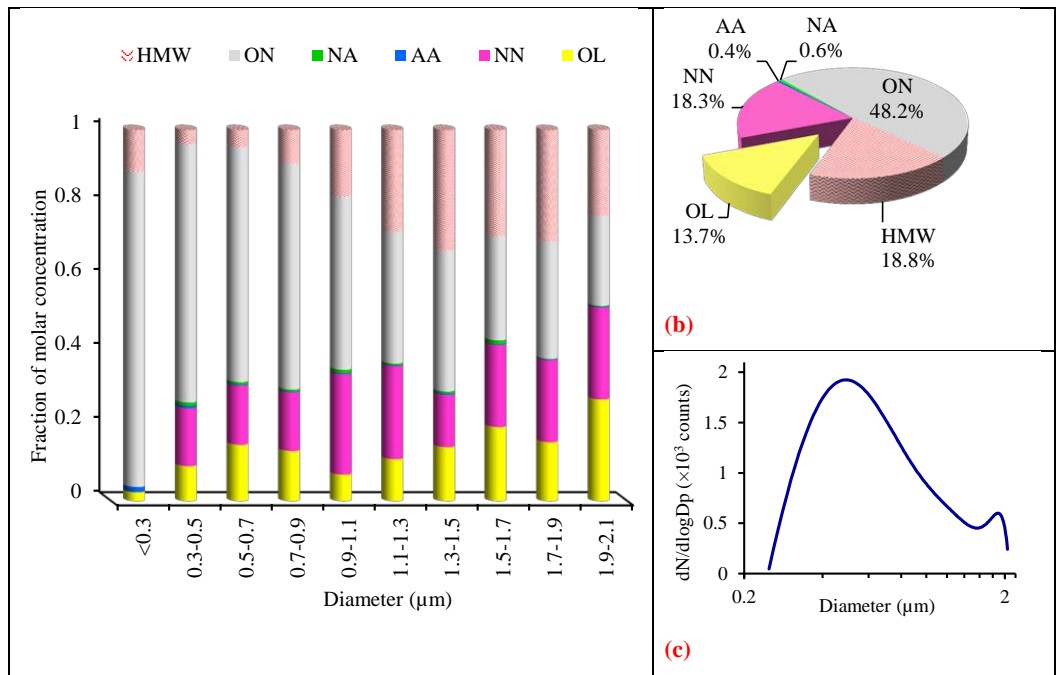

**Figure 7:** ATOFMS data analysis of: (a) mole fraction of ozonolysed OL particles, under dry
conditions, as a function of particle size, (b) total particle composition as given by mole fraction for
all size fractions, and (c) corresponding size distribution of the aged OL aerosol measured by
ATOFMS





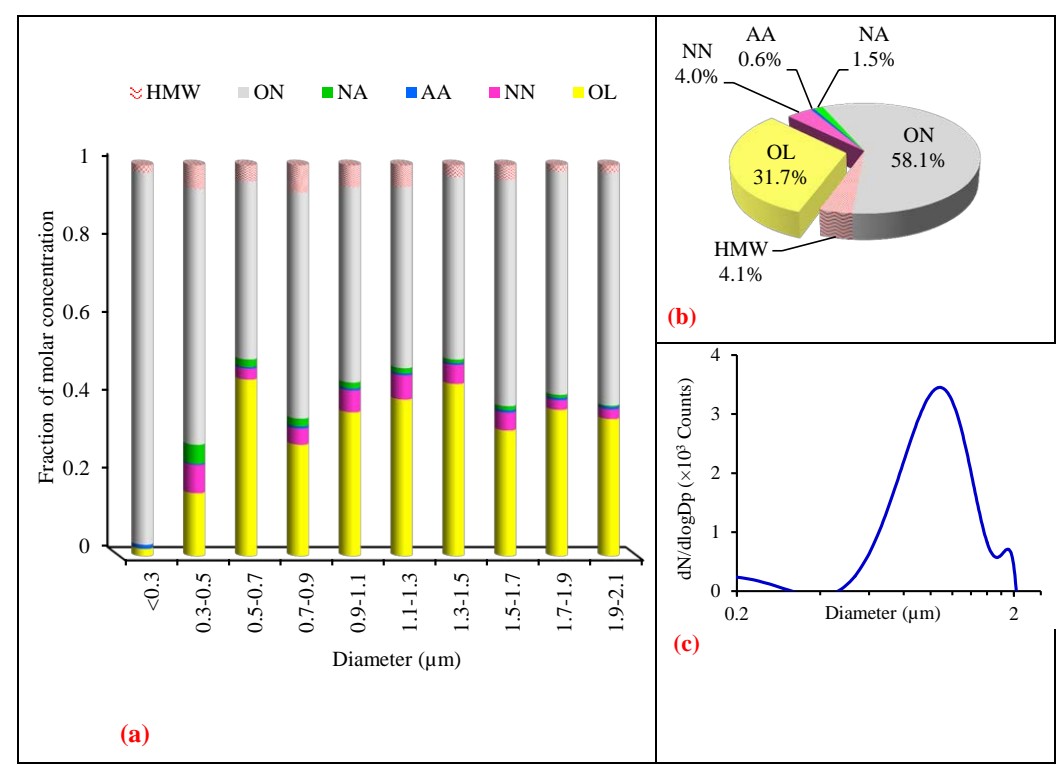

**Figure 8:** ATOFMS data analysis of: (a) mole fraction of ozonolysed OL particles, under
humidified conditions, as a function of particle size, (b) total particle composition as given by mole
fraction for all size fractions, and (c) corresponding size distribution of the aged OL aerosol
measured by ATOFMS

