# Peer review of "SIZE DEPENDENT CHEMICAL AGEING OF"

_Atmospheric Chemistry and Physics, 2016_

## Referee Comment (RC1) · Anonymous Referee #1 · 26 Apr 2016

The manuscript entitled, "Size Dependent Chemical Ageing of Oleic Acid Aerosol under Dry and Humidified Conditions," by Al-Kinki et al. focuses assigning reaction products of the ozonolysis of oleic acid using ATOFMS. Exploring the chemical aging of organic particles relevant in the environment (or as proxy for structural similar systems) and developing new strategies for their quantitative analysis is certainly of fundamental interest. Given the complexity of atmospheric aerosol, laboratory experiments on confined systems are required for developing a better understanding of the complex processes occurring in airborne particles. Oleic acid (OL) + O3 has been extensively studied and there are key elements of the reaction mechanism that are still not understood and may involve the complex evolution of organic diffusion constants with extent of oxidation. The authors present a very detailed experimental description of the apparatus and have gone to impressive lengths to make the ATOFMS mass spectra, to

the extent possible, quantitative. The mass spectral assignments are plausible and the authors present, in most cases, reasonable mechanistic routes for the formation of the high molecular weight products. Since this reaction has been extensively studied, and is a subject of an extensive data evaluation and review paper by Zahardis and Petrucci, the expectation is elevated for any new paper reporting and interpreting results on this reaction in light of the extensive existing literature. The main weakness of the work, which should be addressed by the authors, is the absence of interpreting the observed products within a full kinetic framework (especially in light of previous literature) as detailed below.

1. The connection between the SMPS measurements of particle size (dry and wet) are disconnected from the main theme of the paper which is chemical composition via ATOFMS. The ATOFMS instrument sizes by aerodynamic flight time, so it is not clear what additional information is provided by the SMPS? The authors observed the expected decrease is size previously observed in many other studies. The difference between dry and wet size reduction is insignificant statistically. Therefore, the authors should either better justify the inclusion of the SMPS data and discussion for the main focus of the paper or remove it for brevity.

2. Much of the mechanistic discussion is predicated on the size dependent concentrations shown in Fig. 7 and 8, which doesn't adequately normalize for extent of reaction (# of reactive collisions), since at a single ozone exposure the concentration (even without secondary chemistry) of OL will be size dependent. So any mechanistic conclusions based upon particle size drawn from Fig. 7 and 8 is ambiguous at best. For example, page 23 line 2 it is not unsurprising that larger particles contain more unreacted OL than smaller particles; this is exactly what you would expect kinetically given the difference in the number of OL molecules vs. size and the differences in surface area vs. size. So this observation alone is not sufficient to conclude that limited OL or O3 diffusivity is the cause. This conclusion originally reported by Smith 2002 was drawn only after the full particle size kinetics ([OL] vs. ozone exposure) was measured and

Interactive
comment

reactive uptake coefficients where computed and were then found to be size dependent. Later on page 23 line 13-14 the authors conclude that the molar ratio of O3:OL is smaller for larger particles. . . .in order to explain larger OL for bigger particles in Fig. 7 and 8. Again I don't see how this works since at a fixed ozone concentration larger particles (due to S/V ratio) will always have a less extent of reaction than smaller particles. Furthermore, the authors state that ozone is not appreciably diminished inside their reactor when particles are present. So it is not clear how this small molar ratio of O3:OL works then. Won't the Henry's law constant imply that the [O3] is independent of size? Again on page 24 line 15-16 there is a statement about lower concentration of OL in smaller particles. . . but again this could just arise from a larger extent of reaction for smaller particle than larger ones at a single ozone concentration. It may well be the case that different chemistries are operative in larger particles (secondary chemistry) but such a comparison needs to be done by first eliminating the trivial size dependence due to kinetics. Furthermore, I don't observe any consistent monotonic trend in OL (or ON, NN, HMW) vs. size for example. The authors should address this. The quantity of OL (fig. 7 and 8) oscillates as a function of size, suggesting simply measurement error or uncertainty rather than a robust trend in secondary chemistry/diffusion as a function of size. Some error analysis needs to be shown and included in Fig. 7 and 8.

3. Smith et al. 2002 observe that the uptake coefficient decreases with increasing size. How is this observation reconciled with the present study in which formation of HMW species is observed in large particles, since the mechanisms presented for HMW formation involve the consumption of more than one OL molecule (i.e. secondary chemistry, see R1 and Scheme 2). Thus it would seem from the present results, that larger effective uptakes are expected for larger sized particles (more secondary chemistry since there are additional sinks for OL, rather than just O3) which would be exactly the opposite trend observed by Smith et al. Size dependent kinetic measurements would offer a tangible way to connect the formation of HMW with previous size dependent uptake coefficient measurements of Smith et al.

4. It is not clear to me why, given the experimental setup (ATOMS, SMPS, AFT) why the authors did not measure the OL decay kinetics and product formation kinetics vs. Ozone exposure. This would have addressed #2 above as well as provided needed evidence for some of the proposed reaction mechanisms described throughout the manuscript. This would be entirely new by providing a potential new connection between reactive uptake, particle size, HMW, and diffusive limitations. Furthermore, there should be distinctive kinetic signatures based upon the proposed reaction mechanism. For example, the formation of NN via the decomposition of the primary ozonide (route 1 scheme 2) would be expected to have a different kinetic evolution than NN formed via the reaction of CI1 addition to the OL double bond and subsequent decomposition (as shown in scheme 2). Since the former NN pathway only involves consumption of 1 OL in contrast to the latter which requires for its formation the consumption of 2 OL. This is also true for example in Scheme 3, where AA is formed by CI1 + ON but also could be formed by isomerization CI1 directly.

In summary, the logical arguments about mechanisms based upon difference is OL concentration (and other products) in small and large particles does not necessarily follow, given the way the experiments were performed (all sized measured at a single ozone concentration) and the lack of full kinetic measurements ([OL] vs. Ozone exposure) is needed to compare the same extent of reaction for each particle size. Although it is clear the authors observe some new HMW species whose identification is important for the community, the manuscript needs to be significantly revised for clarity, and further kinetic measurements might be required.

Minor Comments:

1. The split axis, while certainly convenient, makes it very difficult for the reader to ascertain the true relative intensities of the peaks in the mass spectra shown in Figs. 5 and 6. Please reformat these figures with insets (not split axes) of relevant regions so that the reader can, as much as possible, see the raw data.

2. Scheme 3: Elimination of H2O doesn't give the product shown. Do you mean elimination of OH radicals? Intermediates steps are needed to show how H2O is formed.

3. Define what NBS stands for at its first usage page 10 line 17.

4. The ozone concentration is reported at the exit of the generator. Is this the same as the ozone concentration in the flow tube. This should be clarified.

5. The authors should discuss the possibility or not of ion-molecule reactions in the laser ionization region potentially contributing to the mass observed.

6. Check Scheme 2 for consistency regarding upper and lower cases and order of name, mass etc.

7. The first part of the paper, comprising the experimental description, contains a very detailed description of the single experimental components and can be shortened for better readability.

8. P. 9, l 4-6. Redundant, since it won't be referred to at any time in the paper.

9. Panel description (a, b, c) is missing in Figure 2.

10. The comparison of NBS and NBS + OL (Figure S1b) shows next to the OL and OL dimer two additional peaks in the positive spectrum. Can you comment on their origin?

11. Figure 3: change labeling of y-axis. Upper/lower cases in title is inconsistent with other figures, the tick label 0.0 is inconvenient, as is the position of the title. Why are the assignments for the positive ion spectrum of NN and 4-ON missing? Also the OL spectrum is different from the one in the SI, the peak around m/z = 60 is missing, although it does not originate from NBS.

12. Can you make an assumption about the nature of peaks in the positive ion spectrum in Figure 5?

13. Define AAHP (p.21).

14. Panel description (a) is missing in Figure 7.

15. Typographical errors are noted on: p. 2 line 15, 18; p. 4, l. 23; p. 5, l. 2 (2x); p. 7, l. 17, 23; p. 8, l. 16; p. 10, l. 3; p.16, l. 1; p.21, l. 13; p. 25, l 5; p. 41

**[ACPD](ACPD)**

---

## Referee Comment (RC2) · Anonymous Referee #2 · 16 May 2016

Review of Al-Kindl et al., ACPD, 2016

The authors have measured size dependent chemical aging of oleic acid particles under dry and humid conditions. Chemical composition was measured using an ATOFMS instrument, showing that smaller particles contain smaller products such as nonanal and oxononanoic acid, while larger particles contain more high molar mass products. Size distribution was also monitored by SMPS, demonstrating evaporation of nonanal and shrink of oxidized particles. Oleic acid – ozone system has been extensively studies both by experiments and modeling over a decade. The authors did good job in overviewing the past findings in introduction. The experiments seem to be planned carefully and conducted very well. The manuscript is written clearly and it was easy to follow. I am happy to recommend publication of this study in ACP after the below several minor comments are addressed and implemented in the revised manuscript.

[Figure]

Specific comments/questions:

- The actual ozone concentration and reaction time are both not clearly stated. These are critical information to be specified. Ozone concentration seems to be very high (20 ppm at the O3 generator exit) with very short reaction time. This might lead to potential artifacts in ozone uptake and chemical transformation pathways of oleic acid (e.g., Renbaum and Smith, ACP, 11, 6881-6893, 2011). This issue needs to be discussed.

- Was the size distribution under humid condition (presented in Fig. 2b) measured under dry conditions or same humidity as reaction conditions (65% RH) in SMPS? In other words, I am asking whether the sheath flow of SMPS was kept always dry or humidified depending on reaction conditions.

- It seems that exposure (time * O3 concentration) was not large enough to react away all of oleic acid molecules (13-31% of OL remained unreacted in Fig. 7&8). Have you tried to increase exposure? Would particles evaporate even more in that case? Is it possible for authors to present the evolution of particle size (or mass) as a function of reaction time or particle size dependence on O3 concentration?

- P18, L20: Do authors have any evidence to believe it is the primary ozonide, but not secondary ozonide? Is POZ stable enough to be detected by ATOFMS?

- P23, L10: There seems to be misunderstanding of explanation of Shiraiwa et al. (2010). They did not show large concentration gradient of OL in the bulk, but actually they showed OL is homogeneous in the bulk due to rapid bulk diffusion. Ozone can be constrained in the near-surface bulk due to reactions with oleic acid.

- P15, L17: Can you quantify OL dimer? Did it disappear after exposure to ozone? Could it potentially affect reaction kinetics/pathways (e.g., Fig. 9, Zahardis & Petrucci, 2007)?
* * *

---

## Author Comment (AC1) · 30 Sep 2016

Please find attached a response document, including a revised manuscript and SI. This has been uploaded as a PDF document and can be found as an attachment in the Supplement (pdf/zip).

Please also note the supplement to this comment:
http://www.atmos-chem-phys-discuss.net/acp-2016-230/acp-2016-230-AC1-supplement.pdf

---

## Author Comment (AC2) · 30 Sep 2016

**Response to Reviews**

We thank both referees for their review and respond to them point by point below.

**Referee #1**

The manuscript entitled, "Size Dependent Chemical Ageing of Oleic Acid Aerosol under Dry and Humidified Conditions," by Al-Kinki et al. focuses assigning reaction products of the ozonolysis of oleic acid using ATOFMS. Exploring the chemical aging of organic particles relevant in the environment (or as proxy for structural similar systems) and developing new strategies for their quantitative analysis is certainly of fundamental interest. Given the complexity of atmospheric aerosol, laboratory experiments on confined systems are required for developing a better understanding of the complex processes occurring in airborne particles. Oleic acid (OL) + O3 has been extensively studied and there are key elements of the reaction mechanism that are still not understood and may involve the complex evolution of organic diffusion constants with extent of oxidation. The authors present a very detailed experimental description of the apparatus and have gone to impressive lengths to make the ATOFMS mass spectra, to C1 the extent possible, quantitative. The mass spectral assignments are plausible and the authors present, in most cases, reasonable mechanistic routes for the formation of the high molecular weight products. Since this reaction has been extensively studied, and is a subject of an extensive data evaluation and review paper by Zahardis and Petrucci, the expectation is elevated for any new paper reporting and interpreting results on this reaction in light of the extensive existing literature.

The main weakness of the work, which should be addressed by the authors, is the absence of interpreting the observed products within a full kinetic framework (especially in light of previous literature) as detailed below.

1. The connection between the SMPS measurements of particle size (dry and wet) are disconnected from the main theme of the paper which is chemical composition via ATOFMS. The ATOFMS instrument sizes by aerodynamic flight time, so it is not clear what additional information is provided by the SMPS? The authors observed the expected decrease is size previously observed in many other studies. The difference between dry and wet size reduction is insignificant statistically. Therefore, the authors should either better justify the inclusion of the SMPS data and discussion for the main focus of the paper or remove it for brevity.

Response
We believe that the SMPS data provides useful information. Whilst the results were statistically insignificant, there is a suggestion in the data that water might affect the oxidation mechanism "The fact that less mass loss was observed under humidified conditions than under dry conditions (although the difference was statistically insignificant), suggests that the presence of water might have an effect on the oxidation mechanism." The reviewer is correct that the SMPS data is not the main focus of the paper. However, we think it is better placed in the main text rather than the supplementary information.

2. Much of the mechanistic discussion is predicated on the size dependent concentrations shown in Fig. 7 and 8, which doesn't adequately normalize for extent of reaction (# of reactive collisions), since at a single ozone exposure the concentration (even without secondary chemistry) of OL will be size dependent. So any mechanistic conclusions based upon particle size drawn from Fig. 7 and 8 is ambiguous at best. For example, page 23 line 2 it is not unsurprising that larger particles contain more unreacted OL than smaller particles; this is exactly what you would expect kinetically given the difference in the number of OL molecules vs. size and the differences in surface area vs. size. So this observation alone is not sufficient to conclude that limited OL or O3 diffusivity is the cause. This conclusion originally reported by

Smith 2002 was drawn only after the full particle size kinetics ([OL] vs. ozone exposure) was measured and C2 reactive uptake coefficients where computed and were then found to be size dependent. Later on page 23 line 13-14 the authors conclude that the molar ratio of O3:OL is smaller for larger particles. . . .in order to explain larger OL for bigger particles in Fig. 7 and 8. Again I don't see how this works since at a fixed ozone concentration larger particles (due to S/V ratio) will always have a less extent of reaction than smaller particles. Furthermore, the authors state that ozone is not appreciably diminished inside their reactor when particles are present. So it is not clear how this small molar ratio of O3:OL works then. Won't the Henry's law constant imply that the [O3] is independent of size? Again on page 24 line 15-16 there is a statement about lower concentration of OL in smaller particles. . . but again this could just arise from a larger extent of reaction for smaller particle than larger ones at a single ozone concentration. It may well be the case that different chemistries are operative in larger particles (secondary chemistry) but such a comparison needs to be done by first eliminating the trivial size dependence due to kinetics. Furthermore, I don't observe any consistent monotonic trend in OL (or ON, NN, HMW) vs. size for example. The authors should address this. The quantity of OL (fig. 7 and 8) oscillates as a function of size, suggesting simply measurement error or uncertainty rather than a robust trend in secondary chemistry/diffusion as a function of size. Some error analysis needs to be shown and included in Fig. 7 and 8.

Response
On reanalysis of the data we realized we had made a small error in the initial analysis used to average the size resolved data.  We have now replaced the old analysis data with the reanalysed data in the manuscript.  The reanalysis of the data has not changed the conclusions from the paper but does smooth the trend in the size resolved data.  In particular, the major observation of high molecular weight products resulting from the wet oxidation but significantly less in the dry oxidation remains the same.  Figures 7 and 8 have been adapted.

3. Smith et al. 2002 observe that the uptake coefficient decreases with increasing size. How is this observation reconciled with the present study in which formation of HMW species is observed in large particles, since the mechanisms presented for HMW formation involve the consumption of more than one OL molecule (i.e. secondary chemistry, see R1 and Scheme 2). Thus it would seem from the present results, that larger effective uptakes are expected for larger sized particles (more secondary chemistry since there are additional sinks for OL, rather than just O3) which would be exactly the opposite trend observed by Smith et al. Size dependent kinetic measurements would offer a tangible way to connect the formation of HMW with previous size dependent uptake coefficient measurements of Smith et al.

Response
The Lovett et al. paper (2005) which comes from the same research group as Smith et al. (2002) shows that the kinetics of oleic acid ozonolysis is dependent on particle size since it is a surface limited reaction. However, the uptake coefficient remains the same regardless of size.  We observe the same result as Lovett et al. (2005), see response to point 4.

4. It is not clear to me why, given the experimental setup (ATOMS, SMPS, AFT) why the authors did not measure the OL decay kinetics and product formation kinetics vs. Ozone exposure. This would have addressed #2 above as well as provided needed evidence for some of the proposed reaction mechanisms described throughout the manuscript. This would be entirely new by providing a potential new connection between reactive uptake, particle size, HMW, and diffusive limitations. Furthermore, there should be distinctive kinetic signatures based upon the proposed reaction mechanism. For example, the formation of NN via the decomposition of the primary ozonide (route 1 scheme 2) would be expected to have a different kinetic evolution than NN formed via the reaction of CI1 addition to the OL double bond and subsequent decomposition (as shown in scheme 2). Since the former NN pathway only involves

consumption of 1 OL in contrast to the latter which requires for its formation the consumption of 2 OL. This is also true for example in Scheme 3, where AA is formed by CI1 + ON but also could be formed by isomerization CI1 directly.

Response

The aim of this research was not to investigate the kinetics of oleic acid ozonolysis in detail. However, subsequent to the review we have reinvestigated our data sets and we are able to provide several pieces of kinetic information which will be of interest to the reviewer and reader.

We have generated a dataset investigating the kinetics of the ozonolysis of a near-monodisperse sized ensemble of oleic acid particles in the size range of 0.4-0.5 µm with a mean diameter of 0.48 µm. Through the use of 3 different sized aerosol flow tubes (AFT) the interaction time between ozone and oleic acid particles could be obtained at 20, 50 and 135 s. The longest time was the same as that used in the product distribution study shown in Figures 7 and 8. In the absence of $O_3$ the effective interaction time is zero. We observed that the concentration of oleic acid was negligible at the longest interaction time (135 s) and hence the kinetics could only be followed in the time range of 0-50 s. The kinetics of the loss of OL was measured under both dry (0.5% RH) and humidified (65% RH) conditions. No significant difference was observed in dry and wet kinetics which is expected since oleic acid is only marginally hygroscopic and any water that is available will very likely partition to the hygroscopic region of the molecule around the carboxylic acid functional group and not the lipophilic C=C double bond where the ozonolysis occurs. Since the wet and dry runs were very similar, data from both runs were combined into a single dataset to increase the data points available for analysis. The measured data show a linear relationship between the plot of $\ln\{S(OL)/S(OL_0)\}$ versus interaction time, where $S(OL)$ is the oleic acid signal, as shown in figure 9 below. This is consistent with surface limited reaction, as described by Case 3 kinetics initially proposed by Hearn et al. (2005). Using the same approach as Hearn et al. (2005), using the Case 3 kinetics approach, to derive gamma uptake coefficients ($\gamma$), we obtain $\gamma = 5.6\pm0.2 \times 10^{-4}$ which is similar to previous measurements of $\gamma$, e.g. Hearn and Smith (2004), Moise and Rudich (2002), Hearn et al (2005), Thornberry and Abbatt (2004), Ziemann (2005), Knopf et al. (2005).

In addition to the oleic acid reactive decay kinetics, we also observed the time dependent formation of the four major first generation reaction products nonanoic acid (NA), azelaic acid (AA), nonanal (NN) and oxononanoic acid (ON), see Figure 10. The high molecular weight (HMW) products were not observable in these experimental runs; this is consistent with the low levels observed in figures 7 and 8 in the 0.3-0.5 µm size bin. It is noted that low levels of HMW products were observed in the product distribution study but longer averaging times were used. It is clear that whilst the kinetics of oleic acid loss is very similar under both dry and humidified conditions, there are obvious differences in the formation kinetics of the four major first generation reaction products. In particular, under dry conditions the reaction products form more promptly, and once formed stay at relatively similar concentrations. Under humidified conditions the formation of the peak concentration of the products is slower but also their subsequent loss is more substantial. The kinetic data does not provide any definitive mechanistic understanding. However, these results are consistent with the hypothesis that water can act as a reactant with the CI thereby reducing the amount of secondary chemistry observed between OL and the primary reaction products, hence the more stable product distribution after the initial ozonolysis step.

A new section (starting on P27) now details the kinetics of oleic acid ozonolysis.

**Figures**

[Figure]

**Figure 9**. Reactive decay of oleic acid as a function of time for particles in the size range 0.4-0.5 µm (mean diameter = 0.48 µm).  Black circles = measurements, dashed line = linear fit with intercept set to zero.  Both the dry and wet kinetic data has been combined in this plot.

[Figure]

**Figure 10**. Time dependent signals of oleic acid and the four major first generation ozonolysis products.  Graph A was obtained under dry conditions (RH = 0.5%) and B was obtained under wet conditions (RH = 65%).  To aid ease of comparison of the different time series, the signals for all investigated species have been normalized relative to the peak signal achieved by the species investigated.

In summary, the logical arguments about mechanisms based upon difference in OL concentration (and other products) in small and large particles does not necessarily follow, given the way the experiments were performed (all sized measured at a single ozone concentration) and the lack of full kinetic measurements ([OL] vs. Ozone exposure) is needed to compare the same extent of reaction for each particle size. Although it is clear the authors observe some new HMW species whose identification is important for the community, the manuscript needs to be significantly revised for clarity, and further kinetic measurements might be required.

Response
We believe our responses to the major points raised by reviewer 1 answer this last paragraph.

Minor Comments:
1. The split axis, while certainly convenient, makes it very difficult for the reader to ascertain the true relative intensities of the peaks in the mass spectra shown in Figs. 5 and 6. Please reformat these figures with insets (not split axes) of relevant regions so that the reader can, as much as possible, see the raw data.

Response
Figure 5 now has split axis removed and includes an insert panel.

2. Scheme 3: Elimination of H2O doesn't give the product shown. Do you mean elimination of OH radicals? Intermediates steps are needed to show how H2O is formed.

Response
The product shown (M = 342 Da) would be formed by elimination of $H_2O$ from AAHP, in the conversion of the peroxide unit to a carbonyl, although this is likely to be a concerted process with the combination of CI1 and ON, rather than an elementary reaction of the association complex formed from these species. We have amended the text to clarify this.

3. Define what NBS stands for at its first usage page 10 line 17.

Response
NBS is defined as Nile Blue Sulphate at first usage (page 9, line 25)

4. The ozone concentration is reported at the exit of the generator. Is this the same as the ozone concentration in the flow tube. This should be clarified.

Response
Ozone levels were measured at the exit of the aerosol flow tube, not the ozone generator (to take account of losses (negligible) and flow dilution). Chemical consumption of ozone was also found to be negligible. This is now clarified on page 7 (line 15) and page 11 (line 20).

5. The authors should discuss the possibility or not of ion-molecule reactions in the laser ionization region potentially contributing to the mass observed.

Response
The minimal residence time of the ATOFMS instrument (compared with, for example, drift-tube techniques) minimises the scope for within-detector ion-molecule chemistry to contribute to the observed signals; further confidence in this regard is achieved as the bulk of the analysis presented considers *changes* in mass spectrometric signals, upon addition of the chromophore, and particularly

as a function of the AFT chemical reaction conditions. We have added a comment to this effect in the outline of the ATOFMS applicability to this study (page 10).

6. Check Scheme 2 for consistency regarding upper and lower cases and order of name, mass etc.

Response
We have redrawn this scheme for consistency of nomenclature

7. The first part of the paper, comprising the experimental description, contains a very detailed description of the single experimental components and can be shortened for better readability.

Response
We have shortened the experimental description by ca. 1 page. But are keen to retain a sufficiently detailed description to allow repeatability (considering that this is a new experimental set-up).

9. Panel description (a, b, c) is missing in Figure 2.

Response
We have added the description (a,b,c) to the figure.

10. The comparison of NBS and NBS + OL (Figure S1b) shows next to the OL and OL dimer two additional peaks in the positive spectrum. Can you comment on their origin?

Response
As the spectrum shown is a "non-reaction" run these are likely OL fragmentation products. The masses correspond to 61 and 126.

11. Figure 3: change labeling of y-axis. Upper/lower cases in title is inconsistent with other figures, the tick label 0.0 is inconvenient, as is the position of the title. Why are the assignments for the positive ion spectrum of NN and 4-ON missing? Also the OL spectrum is different from the one in the SI, the peak around m/z = 60 is missing, although it does not originate from NBS.

Response
The figure has been redrawn to address the axis label clarity issues.
Mass (m/z) assignments for NN and 4-ON have been added. m/z = 60 had accidently been subtracted from the spectrum when subtracting the NBS signal. This has now been rectified.

12. Can you make an assumption about the nature of peaks in the positive ion spectrum in Figure 5?

Response
These peaks (m/z = 75, 377) do not correspond to any of the first-stage chemical reaction products discussed (or to e.g. simple linear combinations or dimers of these); without further information it is not clear that a definitive assignment can be made.

13. Define AAHP (p.21).

Response
Definition added (page 22 L6)

14. Panel description (a) is missing in Figure 7.

Response
This is added in the revised figure 7.

15. Typographical errors are noted on: p. 2 line 15, 18; p. 4, l. 23; p. 5, l. 2 (2x); p. 7,l. 17, 23; p. 8, l. 16; p. 10, l. 3; p.16, l. 1; p.21, l. 13; p. 25, l 5; p. 41

Response
All typos corrected

**Referee #2**

The authors have measured size dependent chemical aging of oleic acid particles under dry and humid conditions. Chemical composition was measured using an ATOFMS instrument, showing that smaller particles contain smaller products such as nonanal and oxononanoic acid, while larger particles contain more high molar mass products. Size distribution was also monitored by SMPS, demonstrating evaporation of nonanal and shrink of oxidized particles. Oleic acid – ozone system has been extensively studies both by experiments and modeling over a decade. The authors did good job in overviewing the past findings in introduction. The experiments seem to be planned carefully and conducted very well. The manuscript is written clearly and it was easy to follow. I am happy to recommend publication of this study in ACP after the below several minor comments are addressed and implemented in the revised manuscript.

Specific comments/questions:

The actual ozone concentration and reaction time are both not clearly stated. These are critical information to be specified. Ozone concentration seems to be very high (20 ppm at the O3 generator exit) with very short reaction time. This might lead to potential artifacts in ozone uptake and chemical transformation pathways of oleic acid (e.g., Renbaum and Smith, ACP, 11, 6881-6893, 2011). This issue needs to be discussed.

Response
See response to reviewer 1 and alternations to the manuscript outlined therein, which consider potential kinetic and diffusional limitations.  The ozone mixing ratio in the reaction volume (where exposure to OL aerosol occurred) was 20 ppm - we have modified the manuscript (. Total reaction time is clarified (e.g. new figure 10; 20 - 135 seconds)

Was the size distribution under humid condition (presented in Fig. 2b) measured under dry conditions or same humidity as reaction conditions (65% RH) in SMPS? In other words, I am asking whether the sheath flow of SMPS was kept always dry or humidified depending on reaction conditions.

Response
The SMPS sheath flow was derived from the sampled airstream, and so reflected the same humidity.  The experimental timescale (between changes in conditions) was such that ample time was available for the SMPS to relax to changes in RH.

It seems that exposure (time * O3 concentration) was not large enough to react away all of oleic acid molecules (13-31% of OL remained unreacted in Fig. 7&8). Have you tried to increase exposure? Would particles evaporate even more in that case? Is it possible for authors to present the evolution of particle size (or mass) as a function of reaction time or particle size dependence on O3 concentration?

Response
See response to reviewer 1, above, and new manuscript figures 9 and 10, which show essentially complete (within measurement uncertainty) and first-order consumption of OL

P18, L20: Do authors have any evidence to believe it is the primary ozonide, but not secondary ozonide? Is POZ stable enough to be detected by ATOFMS?

Response
There is no experimental evidence, hence our caution in stating this peak *could* be assignable to the POZ. To our knowledge, there are no thermochemical data re the stability of ozonides of this size available, let alone in the OL/NBS particle matrix, so we are reluctant to speculate.

P23, L10: There seems to be misunderstanding of explanation of Shiraiwa et al. (2010). They did not show large concentration gradient of OL in the bulk, but actually they showed OL is homogeneous in the bulk due to rapid bulk diffusion. Ozone can be constrained in the near-surface bulk due to reactions with oleic acid.

Response
We have revised the phrasing (P24 L6) of the manuscript in this section. We are in agreement with the reviewer.

P15, L17: Can you quantify OL dimer? Did it disappear after exposure to ozone? Could it potentially affect reaction kinetics/pathways (e.g., Fig. 9, Zahardis & Petrucci, 2007)?

Response
We cannot quantify the dimer as no standards are available, and the response may differ from (e.g.) OL or the other quantified products - hence we focus upon the kinetics of OL alone. We cannot preclude reactions of the dimer affecting the products formed, and have added a caveat to this effect (page 20 L25).

[revised manuscript text omitted]

---

## Author Response (AR2)

**Journal: ACP Title: Size Dependent Chemical Ageing of Oleic Acid Aerosol Under Dry and Humidified Conditions Author(s): Suad Al-Kindi et al. MS No.: acp-2016-230**

**Dear Editor,**

Error bars have now been added to both figures 9 and 10. Whilst calculating the error bars on figure 9 it was noticed that the wrong data had been graphed in the previous response to reviewers and the quoted gamma was also wrong. We apologize for this mistake which has now been rectified.

The appropriate data set shows some difference in the derived gamma values between the humid and the dry runs. The calculated gammas are as follows:  $\gamma_{dry} = 3.86 \pm 0.54 \times 10^{-4}$ ,  $\gamma_{wet} = 2.40 \pm 0.36 \times 10^{-4}$ , and the value when both the dry and wet data are combined is  $\gamma_{combined} = 3.13 \pm 1.49 \times 10^{-4}$ , where the stated errors are 2× the standard error of the mean. Since only 3 measurements were made under both dry and wet conditions the confidence limits for the combined data set is low, so we choose to show the confidence prediction limits for the combined data set in Figure 9. The new results have not significantly changed our conclusions. The kinetics section has been amended to reflect these changes and is reproduced below. Changes have been highlighted by underlining.

We have also taken on the following editorial comments: "Please make sure to state somewhere in the manuscript that if the reaction occurs at or near the surface, the uptake coefficient remains the same, while the degradation rate of OL is size dependent. You could cite Steimer et al., ACP, 2014, where the math of this has been explicitly outlined. In turn, the size dependent degradation then also allows to draw the conclusion about the kinetic regime (surface reaction or reaction-diffusion limitation), you could cite Berkemeier et al., ACP, 2013, for the definitions and terminology."

In the kinetics section we now state the following – "The observed kinetics are what we expect if the initial reaction between OL and  $O_3$  occurs at or near the surface. Steimer et al. (2016) provide detailed mathematical analysis showing for this reaction scenario the degradation rate of OL is size dependent but the uptake coefficient remains the same. It follows that the kinetics of the ozonolysis of OL is surface limited rather than reaction diffusion limited, detailed discussion of different kinetic regimes are provided in Berkemeier et al (2013)."

**3.11 Kinetics of Oleic acid ozonolysis and product formation**

[revised manuscript text omitted]